# An investigation of causal relationships between prediabetes and vascular complications

Pascal M. Mutie[1,6], Hugo Pomares-Millan [1,6], Naeimeh Atabaki-Pasdar[1], Nina Jordan[2], Rachel Adams [3], Nicole L. Daly[3], Juan Fernandes Tajes[1], Giuseppe N. Giordano [1] & Paul W. Franks [1,4,5✉]

Prediabetes is a state of glycaemic dysregulation below the diagnostic threshold of type 2 diabetes (T2D). Globally, ~352 million people have prediabetes, of which 35–50% develop full-blown diabetes within five years. T2D and its complications are costly to treat, causing considerable morbidity and early mortality. Whether prediabetes is causally related to diabetes complications is unclear. Here we report a causal inference analysis investigating the effects of prediabetes in coronary artery disease, stroke and chronic kidney disease, complemented by a systematic review of relevant observational studies. Although the observational studies suggest that prediabetes is broadly associated with diabetes complications, the causal inference analysis revealed that prediabetes is only causally related with coronary artery disease, with no evidence of causal effects on other diabetes complications. In conclusion, prediabetes likely causes coronary artery disease and its prevention is likely to be most effective if initiated prior to the onset of diabetes.

[1] Genetic and Molecular Epidemiology Unit, Lund University Diabetes Centre, Department of Clinical Sciences, Clinical Research Centre, Lund University, Skåne University Hospital, Jan Waldenströms gata 35, Malmö SE-20502, Sweden. [2] Regulatory Affairs Intelligence, Novo Nordisk A/S, Copenhagen, Denmark. [3] Regulatory Affairs—Neuroscience and Cardiovascular Metabolism, Janssen, High Wycombe, UK. [4] Department of Public Health and Clinical Medicine, Section for Medicine, Umeå University, Umeå, Sweden. [5] Department of Nutrition, Harvard T.H. Chan School of Public Health, Boston, MA, USA. [6]These authors contributed equally: Pascal M. Mutie, Hugo Pomares-Millan. ✉email: paul.franks@med.lu.se

Prediabetes is an impaired state of glucose metabolism defined by elevated but not yet diabetic levels of fasting or 2-h glucose, or HbA1c. The specific cutoffs used to define prediabetes vary but the widely adopted American Diabetes Association (ADA) definitions are: impaired fasting glucose (IFG) = fasting glucose 5.6–6.9 mmol $L^{-1}$; impaired glucose tolerance (IGT) = 2-h glucose 7.8–11.0 mmol $L^{-1}$; HbA1c = 39–46 mmol $L^{-1}$ (or 5.7–6.4%). The cooccurrence of IFG and IGT is termed "impaired glucose regulation".

Whilst the global prevalence of prediabetes in adults is about 7.3% ($n$ = 352 million people), in Europe and the US, roughly 4.6% ($n$ = 36 million people) and 33.9% ($n$ = 84.1 million people) of the adult populations, respectively, are estimated to have prediabetes[1]. In the short term, a relatively small proportion (5–10% annually) of those with prediabetes will progress to full-blown diabetes; however, after 5 years, about half will have developed the disease[2].

As diabetes progresses, it becomes increasingly difficult to treat, as the capacity to endogenously produce insulin diminishes and life-threatening complications arise. About five million people died from diabetes-related complications in 2015, of which more than 50% of the deaths were cardiovascular in nature, with costs attributable to diabetes amounting to about one trillion USD globally as of 2017[1].

Many observational studies have shown that prediabetes is a risk factor for cardiovascular disease (CVD), suggesting that the pathogenic effects of dysregulated glucose metabolism have already begun even before diabetes is manifest[3]. However, these observations cannot be directly interpreted as causal effects owing to the limitations of observational epidemiology. Nevertheless, if prediabetic blood glucose variation was known to cause micro- and/or macro-vascular disease, this could profoundly impact clinical guidelines for the prevention of micro- and macro-vascular disease.

Following a cohort of participants who remain in the prediabetic state for many years would help determine if blood glucose variations within the prediabetic range are associated with CVD; however, such a study is probably unfeasible and would (owing to its observational nature) be prone to confounding and reverse causality. In theory, one could design a clinical trial in which people with prediabetes are randomized to interventions that either (i) maintain blood glucose at the prediabetic level (e.g., by clamping blood glucose and insulin concentrations), or (ii) cause blood glucose control to deteriorate through diabetes and thereafter assess the impact of these interventions on the development of complications. However, for ethical and other pragmatic reasons, such trials are unlikely to be conducted.

Mendelian randomization (MR) is a recently popularized adjunct to randomized controlled trials (RCTs) that makes use of epidemiological data for causal inference. The approach leverages the strengths (stability and random assortment of alleles) of germline DNA variation to generate so-called "instrumental variables" that serve as proxies for environmental exposures[4]. Whilst not without limitations[5], MR is less prone to confounding and reverse causality than observational epidemiology and has been used extensively to validate causal relationships indicated by observational studies.

For the purpose of the current analysis, we have designed an instrumental variable that isolates the exposure of prediabetes from diabetes by selecting single nucleotide polymorphisms (SNPs) with robust signals for variation in nondiabetic glycaemic traits only, with no signal for risk of type 2 diabetes (T2D). We use these instrumental variables to test whether nondiabetic variations in fasting blood glucose (FG) and glycated hemoglobin (HbA1c) are causally related with the most common micro- and macro-vascular complications of diabetes: heart disease, occlusive and hemorrhagic stroke, and renal disease.

## Results

**Observational and MR results.** Thirty-seven articles were included in the meta-analysis of observational studies. The pooled sample size was 1,326,915 participants, with mean (±SD) age 53.2 ± 10.2 years and follow-up duration of 9.6 ± 4.8 years.

In the observational data meta-analysis, prediabetes was associated with a 16% elevated risk of coronary artery disease (CAD) (RR = 1.16; 95% CI: 1.09, 1.23; $Q$ = 52.5, $P_{Qstat}$ = 0.058; $I^2$ = 27.7%; Fig. 1). In the MR analysis, nondiabetic fasting glucose variation was also significantly associated with CAD, such that 1 mmol $L^{-1}$ higher fasting glucose conveyed an OR of 1.26 (95% CI: 1.16, 1.38) for CAD, with no evidence of directional horizontal pleiotropy (Egger intercept = 1, $P$ = 0.76) (Table 1 and Fig. 2). Sensitivity analyses (MR-Egger and weighted median regression) yielded consistent results. Hba1c yielded eight SNPs, which were not classifiable as erythrocytic or glycemic. The association between HbA1c and risk of CAD was not statistically significant (OR = 1.03; 95% CI: 0.64, 1.64) and there was evidence of directional horizontal pleiotropy (Egger intercept = 1.03, $P$ = 0.01; Table 1).

In observational analyses, prediabetes conveyed a RR of 1.11 (95% CI: 1.03, 1.18; $Q$ = 28.5, $P_{Qstat}$ = 0.23; $I^2$ = 16%) for stroke (Fig. 3), these remained virtually unchanged in the subgroup analysis (Supplementary Data 2); however, in the MR analysis, prediabetes was not causally associated with overall stroke (any stroke (AS), OR = 0.88, 95% CI: 0.69, 1.13) or any of the subtypes of stroke (Table 1). Prediabetes was not associated with chronic kidney disease (CKD) in the observational analysis (RR = 1.05; 95% CI: 0.98, 1.12; $Q$ = 27.2, $P_{Qstat}$ = 0.002; $I^2$ = 63.3%), Fig. 4, or in the MR analyses (OR = 1.04; 95% CI: 0.87, 1.25), see below. In the latter, there was no evidence of horizontal pleiotropy.

**Sensitivity analyses.** In further sensitvity and validation analyses of the prediabetes-only instrument, as defined in our study, prediabetes-only SNPs were not significantly associated with T2D risk across all MR methods used, $P$ > 0.05 (Table 2). However, when using all FG SNPs that were genome-wide significant ($P$ < 5 × 10$^{-8}$) regardless of whether or not they were nominally associated with T2D, there was a strong causal relationship between FG and T2D, $P$ < 0.01 across all methods. There was, however, a high degree of horizontal pleiotropy, $P_{Egger\ intercept}$ < 0.01, which underscores the complex nature of T2D (Table 3). All observational pooled estimates remained virtually unchanged in the sensitivity analysis (Supplementary Figs. 1–3).

We further tested for pleiotropy and presence of outliers using the Mendelian Randomization Pleiotropy RESidual Sum and Outlier (MRPRESSO) method for outcomes where outliers were detected—coronary artery disease (CAD), AS and any ischemic stroke (AIS). This method detects horizontal pleiotropy, corrects for it, and also tests the distortion between the corrected and uncorrected causal estimates[6]. The outlier-corrected results did not differ with the inverse-variance weighted (IVW) results for these outcomes (Table 4). In addition, we conducted leave-one-out sensitivity analyses of the relationship between prediabetes and CAD, one using the original 28 SNPs and another using SNPs corrected for outliers using MRPRESSO, to assess whether this association was being driven by one or more influential SNPs. Our results show that the relationship between prediabetes and CAD is not driven by a single (or more) influential genetic variant(s) (Fig. 5). When we used 2-h glucose levels as an instrumental variable for prediabetes, only two SNPs remained after routine quality control (QC) and use of all genome-wide significant SNPs ($n$ = 7 after QC) did not return significant results in association with CAD (Supplementary Note 2 and Supplementary Table 1). Further sensitivity assessments of the relationship between our

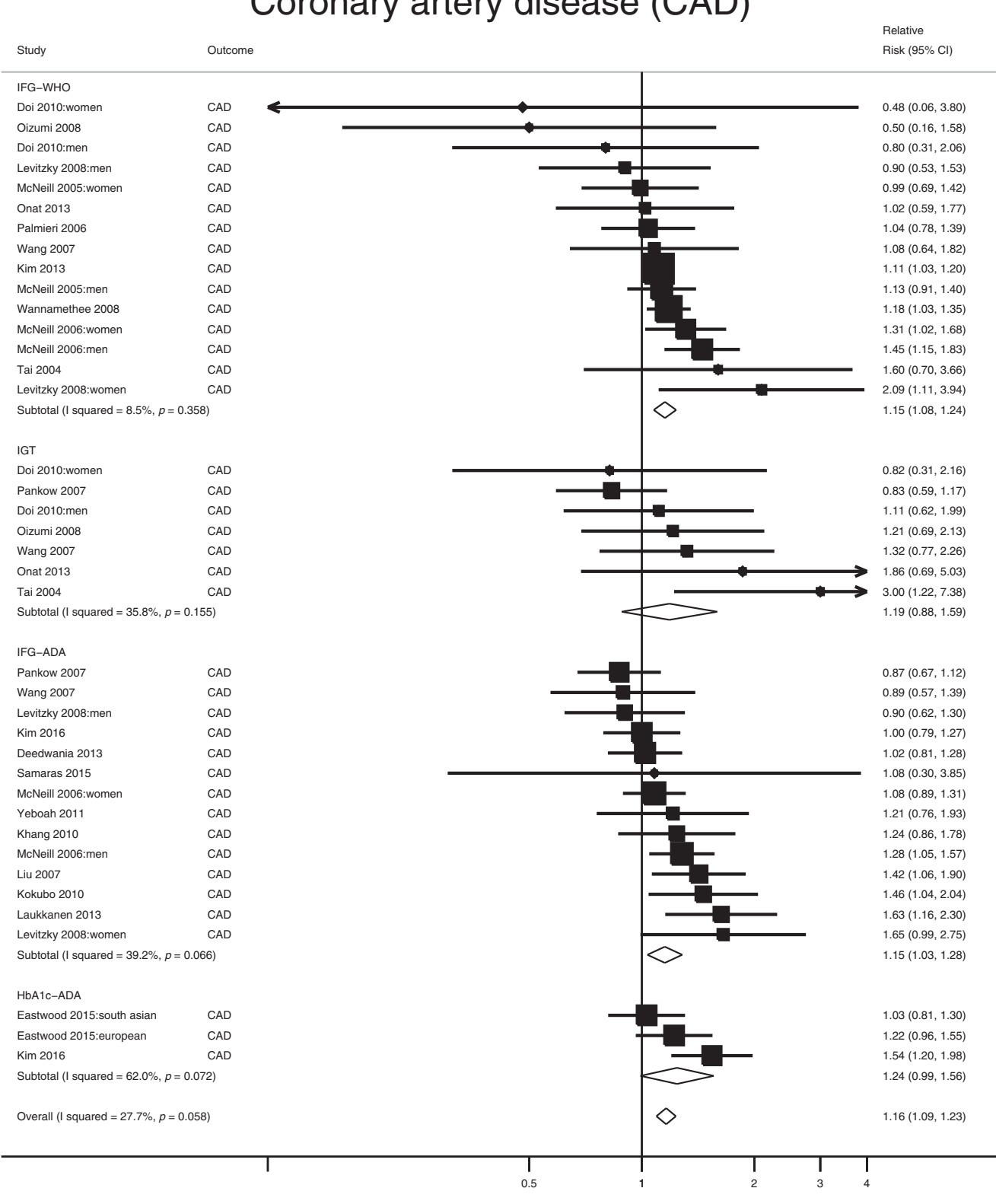

**Fig. 1 Meta-analysis of the association between prediabetes and CAD.** The square and diamond shapes represent effect size (relative risk estimates), while the horizontal bars represent the 95% confidence intervals. A total of 21 studies are included. All *P* values are two-sided. Source data are provided as Source Data file.

**Table 1 Causal relationship between genetically determined prediabetes and vascular outcomes.**

| Trait associated with FG | IVW$_{robust}$ (OR (95% CI)) | MR-Egger (OR (95% CI)) | Egger intercept P value | Weighted median (OR (95% CI)) |
|---|---|---|---|---|
| CAD | 1.26 (1.14, 1.38) | 1.30 (1.09, 1.567) | 0.76 | 1.29 (1.13, 1.47) |
| Any stroke | 0.88 (0.68, 1.13 | 0.71 (0.47, 1.08) | 0.34 | 0.82 (0.64, 1.07) |
| AIS | 0.92 (0.73, 1.16) | 0.70 (0.48, 1.02) | 0.16 | 0.88 (0.67, 1.15) |
| LAS | 0.83 (0.49, 1.40) | 0.66 (0.33, 1.35) | 0.48 | 0.79 (0.43, 1.46) |
| CES | 1.10 (0.75, 1.63) | 0.79 (0.39, 1.58) | 0.21 | 1.04 (0.63, 1.73) |
| SVS | 0.78 (0.46, 1.31) | 0.49 (0.19, 1.22) | 0.23 | 0.61 (0.33, 1.11) |
| CKD | 1.04 (0.87, 1.25) | 0.83 (0.56, 1.22) | 0.32 | 0.93 (0.75, 1.16) |
| HbA1c-CAD[a] | 1.03 (0.64, 1.64) | 0.17 (0.04, 0.79) | 0.01 | 0.83 (0.53, 1.31) |

Data are presented as odds ratios and 95% CI for three methods of the Mendelian randomization analysis. Source data are provided as Source Data file.
*IVW* inverse-variance weighted, *CAD* coronary artery disease, *AIS* any ischemic stroke, *LAS* large artery stroke, *CES* cardioembolic stroke, *SVS* small vessel stroke, *CKD* chronic kidney disease.
[a]Two-sample MR results of the association between genetically determined HbA1c levels and CAD using robust IVW.

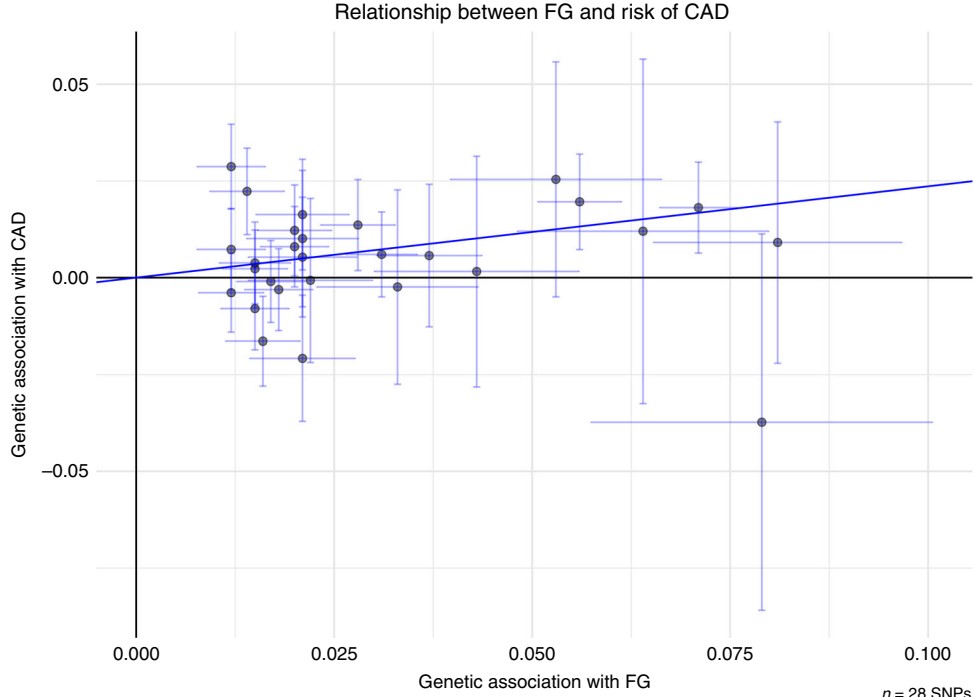

**Fig. 2 Relationship between genetic effects of prediabetes only and CAD.** Data are represented as log-odds and 95% confidence intervals for each trait. Slope of the line represents an estimate of the causal effect of fasting glucose on risk of CAD. The points represent effect sizes for each individual genetic variant (SNPs) for each of the traits on both axes. The horizontal and vertical bars at each point represent the 95% confidence intervals for genetic associations with FG and CAD, respectively. FG fasting glucose, CAD coronary artery disease. Source data are provided as Source Data file.

prediabetes instruments and other cardiovascular risk factors (Total, LDL, and HDL cholesterol levels; tryglyceride levels; and body mass index) did not show any significant association (Supplementary Note 2 and Supplemetary Tables 2–6).

## Discussion

It is unclear if prediabetes is pathogenic or merely a prelude to the disease state of diabetes. We sought to address this important question using MR to estimate the causal effect of nondiabetic variations in FG on the major complications of diabetes. We compared these findings with those obtained through meta-analysis of published observational data from 1,326,915 participants. In the observational analysis, prediabetes was modestly associated with CAD and stroke, but not with CKD. In the MR analyses however, only prediabetic blood glucose was associated with CAD, with a 26% higher odds of CAD per mmol L$^{-1}$

increase in fasting glucose. Elevation in genetically determined HbA1c did not confer a statistically significant increase in the odds of CAD or any other outcomes, though the number of instruments was less ($n = 8$) and the instruments were unclassifiable.

To date, there has been no medicinal products approved for the treatment of prediabetes in the EU or US. While lifestyle measures are clearly recommended as first-line intervention to improve glycaemia in people at high risk of developing diabetes, it is widely acknowledged that additional drug therapy may be beneficial in people with prediabetes, if their risk of diabetes is elevated for other reasons.

Current regulatory requirements for supportive evidence include showing that delay in disease progression is accompanied by other indicators of clinical benefit[7]. To provide this evidence, large, long-term clinical trials are needed, the high cost of which inhibits the development of prediabetic medicinal products. Moreover, there are reimbursement challenges of treating very

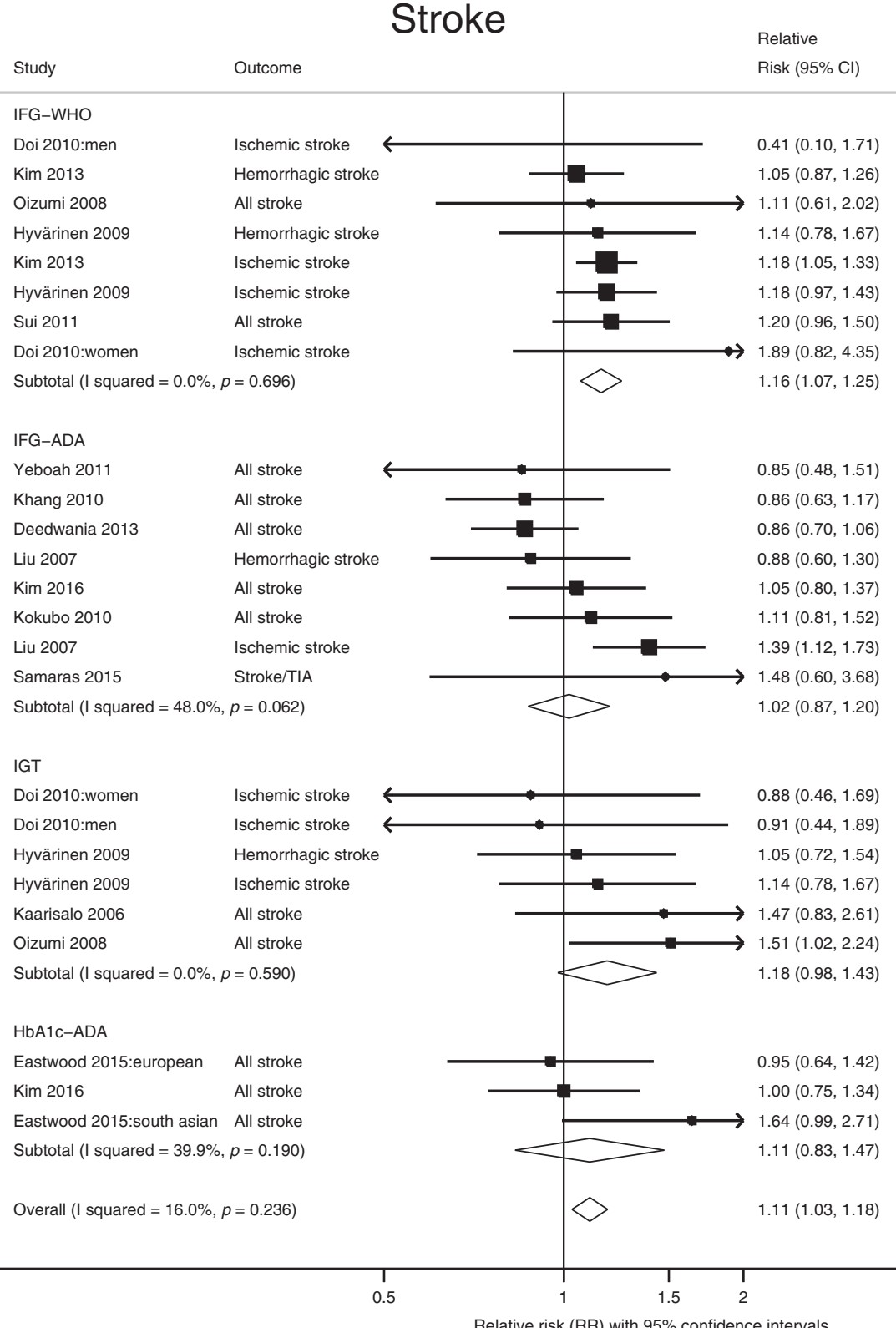

**Fig. 3 Meta-analysis of the association between prediabetes and stroke.** The square and diamond shapes represent effect size (relative risk estimates), while the horizontal bars represent the 95% confidence intervals. A total of 14 studies are included. All *P* values are two-sided. Source data are provided as Source Data file.

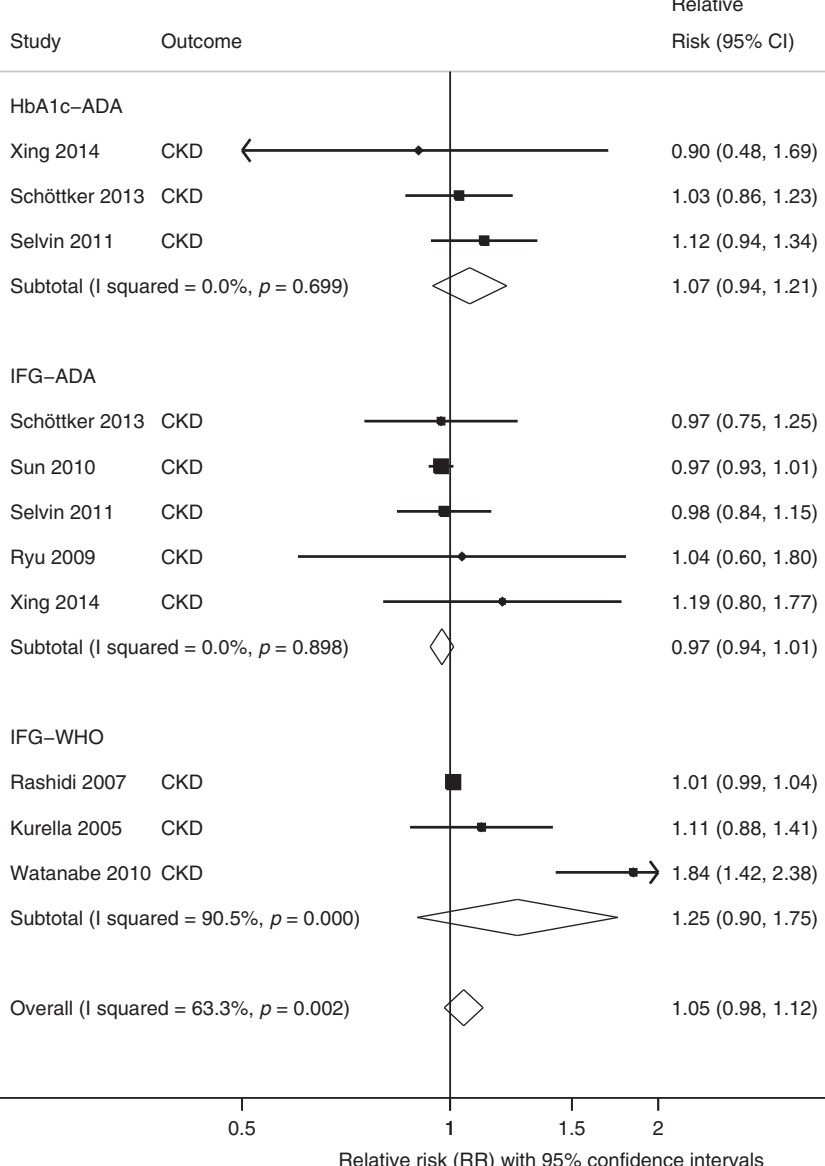

**Fig. 4 Meta-analysis of the association between prediabetes and CKD.** The square and diamond shapes represent effect size (relative risk estimates), while the horizontal bars represent the 95% confidence intervals. In total, eight studies are included. All *P* values are two-sided. Source data are provided as Source Data file.

**Table 2 Causal association between prediabetes only and risk of T2D.**

| Method | OR | Lower 95% CI | Upper 95% CI | P value |
|---|---|---|---|---|
| Weighted median | 0.98 | 0.82 | 1.14 | 0.79 |
| IVW | 1.02 | 0.90 | 1.16 | 0.76 |
| Robust IVW | 1.02 | 0.90 | 1.15 | 0.77 |
| MR-Egger | 0.91 | 0.73 | 1.14 | 0.42 |
| Intercept$_{MR-Egger}$ | 1.00 | 1.00 | 1.01 | 0.23 |
| Robust MR-Egger | 0.91 | 0.77 | 1.07 | 0.25 |
| Intercept$_{Robust\ MR-Egger}$ | 1.00 | 1.00 | 1.01 | 0.15 |

*n* = 28 SNPs. Results are from two-sample Mendelian randomization analyses and *P* values are two-sided. Results are unadjusted for multiple comparisons. Source data are provided as Source Data file.
*IVW* inverse-variance weighted, *OR* odds ratio.

large numbers of people with prediabetes. Determination of the health implications and risk assessment of prediabetes would, therefore, aid design of smaller, shorter, and potentially less expensive, clinical trials by providing alternative health benefits. It would also help address the value of treating large populations over longer periods, by showing cost effectiveness.

MR is often considered an analogue of RCTs. In the latter, treatment allocation is randomized to help ensure that any potential confounding factors that exist within the cohort prior to treatment assignment are distributed evenly between treatment arms, thus neutralizing their impact. In MR analyses, germline DNA variants are used as proxies (instrumental variables) for the exposure of interest (in this case, prediabetes). The random assortment of alleles during meiosis and the stability of DNA variants across the lifespan reduce to a bare minimum the possibility that the observed effect of the instrumental variable

**Table 3 Causal association between fasting glucose (all GWA significant) and risk of T2D.**

| Method | OR | Lower 95% CI | Upper 95% CI | P value |
|---|---|---|---|---|
| Weighted median | 1.55 | 1.23 | 1.94 | $1.67 \times 10^{-4}$ |
| IVW | 2.26 | 1.37 | 3.74 | $1.43 \times 10^{-3}$ |
| Robust IVW | 2.35 | 1.50 | 3.67 | $1.75 \times 10^{-4}$ |
| MR-Egger | 0.46 | 0.19 | 1.12 | 0.09 |
| Intercept$_{\text{MR-Egger}}$ | 1.05 | 1.03 | 1.08 | $5.05 \times 10^{-5}$ |
| Robust MR-Egger | 0.96 | 0.45 | 2.03 | 0.91 |
| Intercept$_{\text{Robust MR-Egger}}$ | 1.03 | 1.01 | 1.04 | $5.54 \times 10^{-3}$ |

$n = 74$. Results are from two-sample Mendelian randomization analyses and $P$ values are two-sided. Results are unadjusted for multiple comparisons. Source data are provided as Source Data file. *IVW* inverse-variance weighted, *OR* odds ratio.

**Table 4 MRPRESSO analysis of relationship between prediabetes and outcomes with detected outliers.**

| Outcome | MR analysis | OR (95% CI) | P value |
|---|---|---|---|
| Coronary artery disease | Raw | 1.27 (1.09, 1.47) | $4.9 \times 10^{-3}$ |
| | Outlier-corrected | 1.24 (1.12, 1.38) | $5.8 \times 10^{-4}$ |
| Any stroke | Raw | 0.92 (0.73, 1.17) | 0.51 |
| | Outlier-corrected | 0.90 (0.72, 1.11) | 0.32 |
| Any ischemic stroke | Raw | 0.95 (0.75, 1.22) | 0.71 |
| | Outlier-corrected | 0.90 (0.74, 1.09) | 0.28 |

All $P$ values are two-sided. "Raw" refers to original FG SNPs ($n = 28$). Source data are provided as Source Data file.
*OR* odds ratio, *CI* confidence interval.

on the outcome is confounded or attributable to reverse causality[4].

Here, we specifically sought to isolate the causal effects of prediabetes from those of diabetes by selecting variants that are robustly associated with fasting glucose and HbA1c variation but not with diabetes. It is hard to envisage a clinical trial where this could be recapitulated, as participants would need to be exposed to prediabetes without progressing to diabetes long enough for complications to occur. Consider, too, that the method used to maintain the prediabetic state would need to function without directly affecting the trial's outcomes, excluding virtually all known blood glucose therapeutics. Thus, for this specific research question, MR is an especially powerful method for causal inference.

One of few naturally occurring examples where blood glucose can remain in the prediabetic state for long periods is a rare form of monogenic diabetes (MODY2), caused by mutations in the glucokinase gene (*GCK*). In MODY2, the blood glucose set-point is elevated, but is generally not linked with progressively deteriorating glycemic control. Moreover, most MODY2 patients do not develop macro- and micro-vascular complications[8]. As intriguing as this is, the physiological idiosyncrasies of the disease limit inferences about vascular risk in prediabetes. For example, unlike many people with prediabetes, MODY2 patients have normal post-prandial glycemic responses, virtually no insulin resistance and cardioprotective lipid profiles[9].

Although this is the first study to our knowledge to undertake a comprehensive systematic literature review coupled with a detailed MR analysis to specifically examine the causal effects of prediabetic blood glucose variation in micro- and macro-vascular disease, previous studies have examined the cardiogenic effects of diabetic and nondiabetic blood glucose variations. In general, the findings from these studies support the clinical consensus that T2D causes heart disease[10].

At least one previous MR study examined fasting glucose variation (inclusive of diabetes) in ischemic stroke and found no statistically robust evidence of effect[11]. However, a published MR analysis that, like our study, harnessed genetic variants associated with glucose but not diabetes[12], also reported evidence of causal associations with CAD. Another measure of glycemia, HbA1c, which reflects average glucose levels over the preceding 3 months, was shown in a recent study to be causally associated with cardiovascular complications[13]. However, as shown here, these results may not be independent of the effects of fasting glucose in CVD.

MR is not without limitations. Canalization is a widely described caveat of MR analyses; the phenomenon occurs when genetic perturbations are offset by coexisting and compensatory mechanisms, effectively short-circuiting the exposure-outcome relationships that MR analyses seek to assess[4]. There are no established methods to detect canalization in MR analyses. Canalization could invalidate MR findings by altering the effect of the genetic instrument on the outcome of interest without affecting the association between genotype and exposure of interest[4]. There are other established methodological limitations of MR, such as horizontal pleiotropy and population stratification, which were overcome in the current analysis using established statistical solutions. A further important consideration is that the exposures characterized in MR experiments should be viewed as having lifelong effects, whereas the timeframe for prediabetes exposure will be confined to a much shorter duration. Thus, the estimated effect of prediabetes in CAD derived from our MR analysis may be greater in magnitude than one would observe in the real world. However, the results from our observational meta-analysis are largely consistent with our MR estimates.

A major limitation of observational studies is the potential that participants progress to diabetes. Therefore, we went to great lengths to identify and stratify those studies which excluded individuals with diabetes in the analysis. Those which we deemed having the most likelihood of enrolling diabetics (i.e., those recruiting participants only with HbA1c or fasting glucose) were further stratified into a specific subgroup for re-analysis; results remained virtually unchanged (see Supplementary Material 2, Table 1, subgroup analysis). By no means do we claim that the observational evidence is definitive; on the contrary, this motivated us to contest these observational data and explore causality through the MR approach.

In conclusion, we report the synthesis of a very large body of epidemiological evidence linking prediabetes with the life-threatening complications caused by diabetes and validate these findings using MR. We found that prediabetes is likely to be causal in CAD, whereas it is not likely to cause kidney disease or stroke. The major implication of this finding is that interventions for the prevention of diabetes-related CAD may be more effective

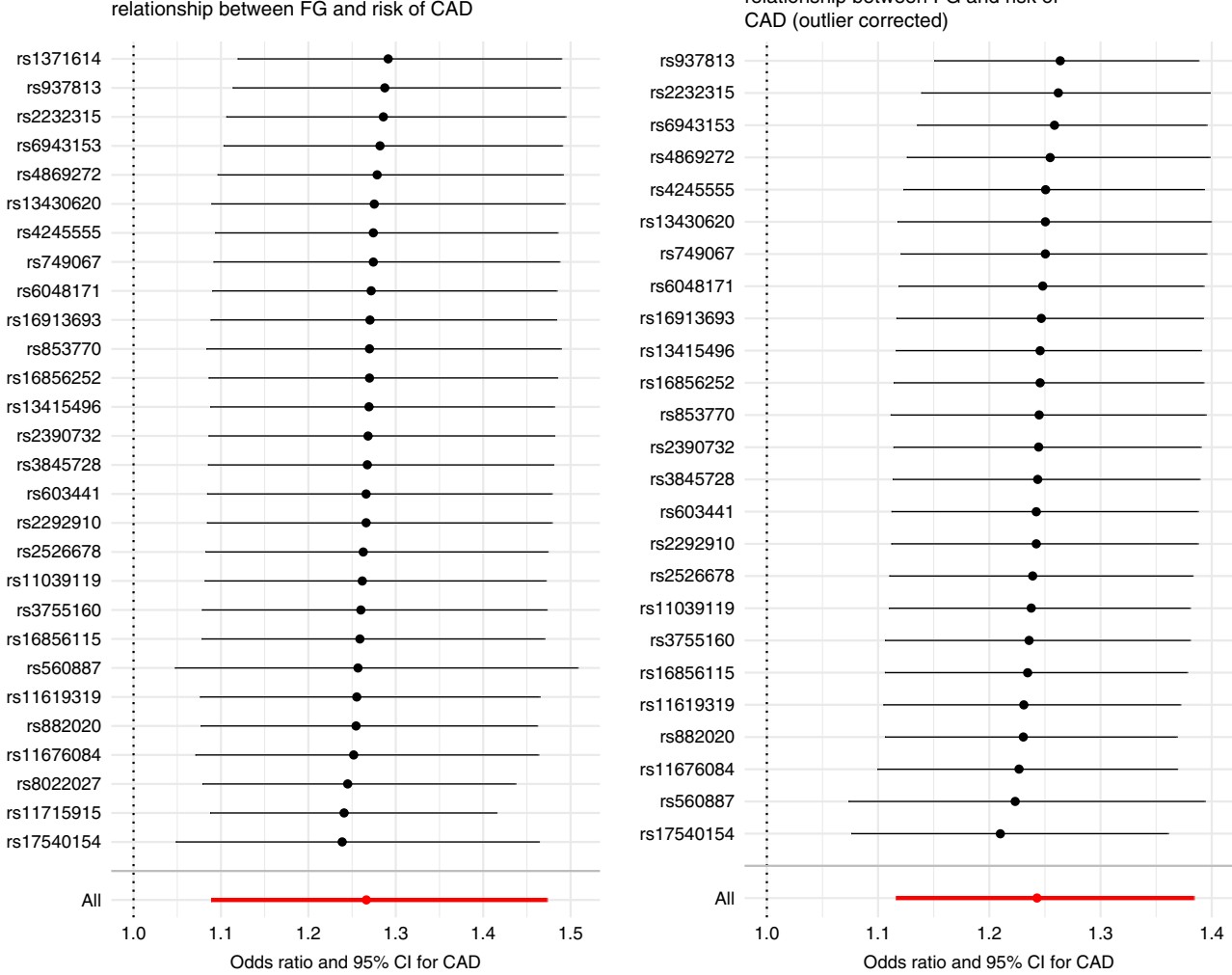

**Fig. 5 Leave-one-out analysis plots of causal relationship between fasting glucose and CAD.** Data are presented as odds (OR) ratio and 95% confidence interval (95% CI) of the exposure-outcome relationship for each SNP. Center points represent the causal effect estimate and the horizontal bars represent the respective 95% CI. Left panel represents data from all SNPs that passed QC ($n = 28$) while right panel represents SNPs retained after correcting for outliers using MRPRESSO, $n = 25$ SNPs. Source data are provided as Source Data file.

if initiated prior to diabetes onset. This may also help explain why CAD prevention in people with established diabetes has proven extremely challenging[14].

## Methods

**Observational data meta-analysis**. We first performed a systematic literature review of published epidemiological studies focusing on "prediabetes and diabetic complications" and extracted summary statistics that we, thereafter, combined through meta-analysis. We then tested the hypothesis that these observational associations were of a causal nature using MR and compared effect estimates derived from the observational meta-analysis and the MR analyses.

A combined medical subject headings term and text search strategy was formulated restricted to "humans" and English language articles (Supplementary Data 1 shows the search strategy in detail). A search of the electronic database PubMed was carried out for all cohort studies published through November 30th, 2017, according to the following criteria: prediabetes defined by IGT, IFG per WHO[15] or ADA criteria, and glycated hemoglobin (HbA1c) per ADA criterion[16]. Studies were included if participants were drawn from the general population, glycaemia was measured at baseline, and the subsequent outcomes at follow-up were CAD, CKD, or stroke, and were compared with the group of normoglycaemic participants. Studies with individuals known to be diagnosed with diabetes or with diabetic values at baseline or follow-up were excluded from the analysis. Figure 6 shows the study selection procedure.

Data extraction: two authors (H.P.-.M. and P.M.M.) independently identified, screened, and reviewed for eligibility the papers identified using the approach defined above. We systematically abstracted data relating to: author(s), year

published, country or region, prediabetes definition, prevalence (%), sample size, gender ratio of the study population (%), participants' age, duration of follow-up, glycaemic status at baseline, outcome definition and ascertainment, covariates and approach used to control for confounding, risk estimates and 95% confidence intervals, in a standard form (Supplementary Data 2 shows the studies' characteristics). Discrepancies in study identification were adjudicated by a third researcher (G.N.G.). Quality of the studies and bias assessment was determined using the Newcastle–Ottawa scale[15] (Supplementary Data 2). Reported findings by subgroups (i.e., sex or ethnicity) were included separately by strata for statistical analysis. Effect estimates (relative risk, hazard ratio, and odds ratio, converted to RR) were logarithmically transformed and standard errors calculated[16]. A priori, we assumed there would be heterogeneity across the cohorts given the differences in population characteristics, follow-up duration, research methods, and outcome definitions. Therefore, the DerSimonian and Laird random-effects model for meta-analysis was used, which is considered more conservative than fixed-effect models[16]. Heterogeneity between and within studies was explored through subgroup analysis (Supplementary Data 2).

Publication bias was assessed using funnel plots and the Begg's and Egger's test. Sensitivity analysis was carried out by omitting one study at a time. All statistical meta-analyses were undertaken with the software Stata 13.0 (Stata Corp LP, College Station, TX).

**MR analyses**. MR is a method that employs instrumental variables to assess the causal association between a given exposure and an outcome[4]. For an instrument to be valid, it must mediate its effect on the outcome only through the exposure and not via other pathways. Further, it should only be associated with the exposure and not be associated with cofounders of the exposure-outcome association[17]. To

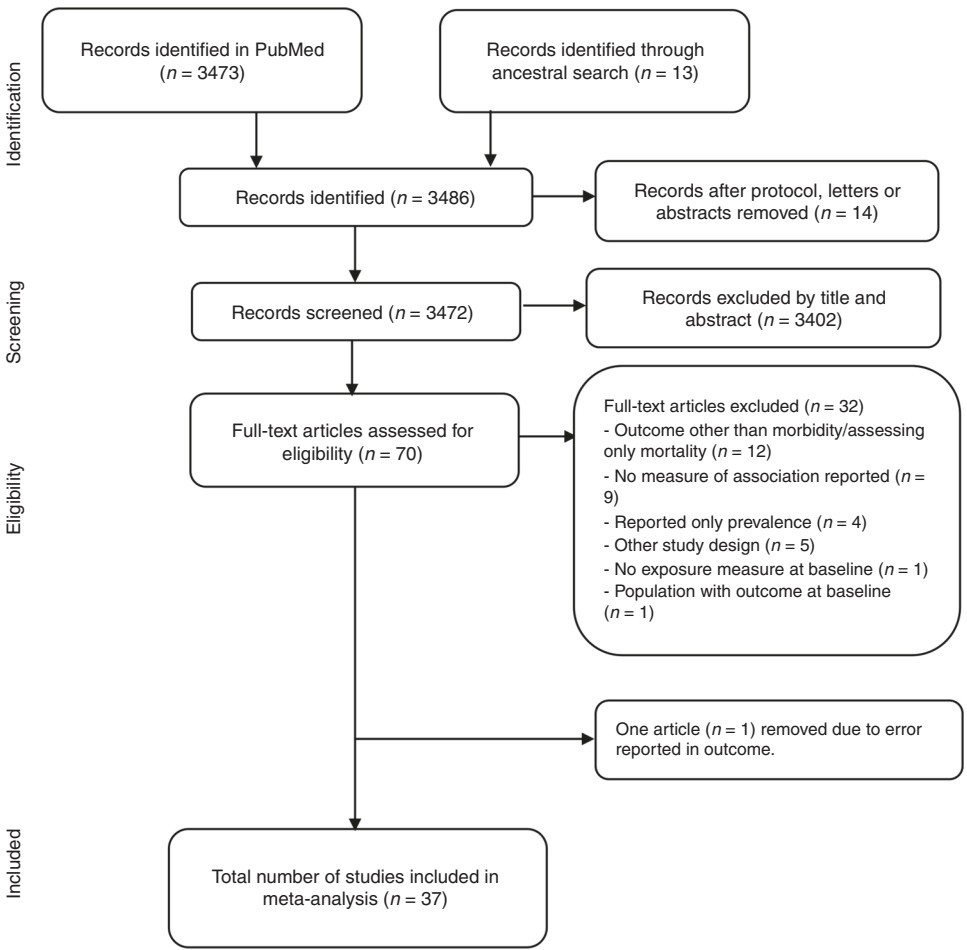

**Fig. 6 Outline of study selection procedure.** Source data are provided as Source Data file.

reduce potential bias due to population stratification, we restricted MR analyses to participants of European descent.

We defined two sets of instruments that specifically characterized variations in fasting glucose and HbA1c within the nondiabetic range. We achieved this by selecting SNPs that are associated with fasting glucose and HbA1c at a genome-wide level of statistical significance ($P < 5 \times 10^{-8}$) within the most recent MAGIC database[18,19], but which are not associated with type 1 or T2D ($P > 0.05$) in the most recent release of the Diabetes Genetics Replication and Meta-analysis database[20,21]. The sets of instruments derived from these variants were then examined within GWAS databases for any respective "diabetic" complications. Specifically, we used publicly available GWAS meta-analysis summary statistics from various consortia. Fasting glucose (exposure) data were obtained from the Meta-Analyses of Glucose and Insulin-related traits Consortium (MAGIC, $n = 133,010$ for fasting glucose)[22]. The MAGIC GWAS meta-analysis includes 32 cohorts, which comprised participants of European descent adjusted for age and sex. Fasting glucose was expressed in mmol L$^{-1}$ and was untransformed in the analyses[18].

HbA1c (exposure) data were also obtained from the latest MAGIC transethnic genome-wide association meta-analysis of genetic variants associated with HbA1c. This meta-analysis included 159,940 participants from 82 cohorts of different ancestries (European, South and East Asian, and African). Individuals of European ancestry were the majority, about 120,962 across 55 cohorts. All participants were diabetes free and studies reported HbA1c as percentage[19].

CAD GWAS summary statistics were obtained from the latest cardiomics meta-analysis data repository[23]. This data comprised of 34541 cases of CAD and 26,1984 controls from the UK Biobank and replication was done in 88,192 cases and 162,544 controls from Coronary Artery Disease (C4D) Genetics consortium (CARDIoGRAMplusC4D)[24,25].

Summary statistics for five phenotypes of stroke (AS, AIS, large artery stroke, cardioembolic stroke, and small vessel stroke) were obtained from the most recent MEGASTROKE consortium meta-analysis data repository[26] in which the analysis for European only ancestry consisted of 40,585 cases and 406,111 controls[27].

Data on renal disease were obtained from the CKDGen GWAS summary data repository[28]. GWAS meta-analysis for CKD (defined as eGRFcrea <60 ml per min per 1.73 m$^2$) was performed on a sample of 745,348 and replicated in a sample of 280,722 giving a combined sample size of more than one million[29].

Selection of glucose-associated SNPs from MAGIC[30], as outlined above, resulted in 47 SNPs for fasting glucose and 10 for HbA1c that we considered reflective of prediabetic glucose variation. To rule out linkage disequilibrium (LD) between SNPs, we performed LD-clumping restricted to $r^2 < 0.2$, a 1000 kb window and retained SNPs with the lowest $P$ value resulting in final sets of 28 uncorrelated fasting glucose SNPs and 8 HbA1c SNPs. For each outcome, these genetic variants were further validated for use in the final analysis. Specifically, the exposure-outcome datasets were harmonized to ensure the same number of SNPs in exposure and outcome sets, similar strand orientation, correct direction of effect sizes, and correcting for palindromic SNPs[31].

**Statistical analysis.** All MR analyses were conducted with the R statistical software v3.6.1 using the MendelianRandomization[32] and TwoSampleMR packages[33].

We used the robust IVW method for the main analysis and the robust MR-egger and weighted median methods for sensitivity analyses. IVW is a widely-accepted approach for MR analyses, which involves regressing the effect sizes of the SNP-outcome association on the SNP-exposure association with the inverse of the variance used as weights. In robust regression, extreme values are penalized to minimize bias.

MR-Egger is used to test for directional horizontal pleiotropy, a violation of the instrumental variable assumption where the effect of the instrumental variable on the outcome is mediated via another pathway other than the exposure of interest. MR-Egger tests for violation of IV assumptions and bias in the inverse variance-weighted (IVW) methods and includes the intercept as part of the regression (unlike IVW, where the intercept is forced to zero)[34]. The resulting coefficient, therefore, provides an asymptotically consistent estimate of the causal effect, even if all variants are pleiotropic with the outcome[35]. This holds when the Instrument Strength Independent of Direct Effect assumption is true, i.e., the instrument strength is independent of its pleiotropic effect. When this criterion is met, MR-Egger provides an unbiased assessment of the association between the exposure and outcome, providing the intercept, which provides the average pleiotropic effect, does not significantly differ from the null. When the intercept is significantly different from the null, it represents an estimate of the directional horizontal pleiotropic effect of the genetic variants[35]. The median-weighted method provides

a reliable estimate of the causal association between exposure and outcome when at least half of the instrumental variables are valid[36].

**Sensitivity analyses and instrument validation**. To rule out false positive associations, we conducted sensitivity analyses to further test the veracity of our instrumental variables. First, we tested the association between the prediabetes instruments with T2D to demonstrate that our instruments represented prediabetes only and rule out any pleiotropic relationship with T2D. Second, we tested the association between all fasting glucose SNPs that reached GWA significance ($n = 74$ after QC) and the risk of T2D, to cement the above facts. Further, we tested if there was any causal relationship between fasting glucose and other cardiometabolic risk factors i.e., BMI, cholesterol levels (total, LDL, and HDL), and triglyceride levels. We also additionally used MRPRESSO to test for horizontal pleiotropy and outliers[6].

**Reporting summary**. Further information on research design is available in the Nature Research Reporting Summary linked to this article.

## Data availability

The GWAS summary statistics data analyzed here are available in the following public repositories. CAD (Dataset: CAD_META.gz): https://data.mendeley.com/datasets/gbbsrpx6bs/1#file-67c31537-5906-40bb-9820-8764b1554666 (https://doi.org/10.17632/gbbsrpx6bs.1)[23]. CKD (Dataset: CKD overall European ancestry): http://ckdgen.imbi.uni-freiburg.de/[28]. T2D (Dataset: T2D GWAS meta-analysis—Unadjusted for BMI[20]): https://www.diagram-consortium.org/downloads.html[21]. Fasting glucose, 2-h glucose, and HbA1c: https://www.magicinvestigators.org/downloads/[22]. The fasting and 2-h glucose datasets are filed under Metabochip replication datasets, and the zipped file contains both datasets (ftp://ftp.sanger.ac.uk/pub/magic/MAGIC_Metabochip_Public_data_release_25Jan.zip). The HbA1c dataset can be retrieved at ftp://ftp.sanger.ac.uk/pub/magic/HbA1c_METAL_European.txt.gz. Stroke: https://megastroke.org/download.html[26]. The dataset (MEGASTROKE_data.zip) is accessible after agreeing to terms of use and submitting a brief project description. Lipids: http://csg.sph.umich.edu/willer/public/lipids2013/[37]. The datasets are filed under "RESULT FILES," subheading "JOINT ANALYSIS OF METABOCHIP AND GWAS DATA." The names of the files are LDL Cholesterol, HDL Cholesterol, Triglycerides, and Total Cholesterol. Body mass index: http://portals.broadinstitute.org/collaboration/giant/index.php/GIANT_consortium_data_files[38]. The dataset is filed under "BMI and Height GIANT and UK BioBank Meta-analysis Summary Statistics." The name of the file is "Meta-analysis Wood et al. + UKBiobank 2018 GZIP". Source data are provided with this paper.

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

## Acknowledgements
We extend our gratitude to the many research groups that have made GWAS summary statistics data publicly available and accessible to the rest of the research community and all participants involved in the numerous studies. This project has received funding from the Innovative Medicines Initiative 2 Joint Undertaking under Grant agreement No. 115881 (RHAPSODY). This Joint Undertaking receives support from the European Union's Horizon 2020 Research and Innovation Programme and EFPIA. RHAPSODY is also supported in part by the Swiss State Secretariat for Education, Research and Innovation (SERI) under contract number 16.0097. This study also received support from the Swedish Research Council, Strategic Research Area Exodiab, (Dnr 2009-1039), the Swedish Foundation for Strategic Research (IRC15-0067), the Swedish Research Council, Linnaeus Grant (Dnr 349-2006-237), and the European Research Council (CoG-2015_681742_NASCENT). The MEGASTROKE project received funding from sources specified at http://www.megastroke.org/acknowledgments.html. The opinions expressed and arguments employed herein do not necessarily reflect the official views of these funding bodies.

## Author contributions
P.M.M.: literature search, data analysis, data interpretation, and writing of the manuscript; H.P-.M.: literature search, data analysis, data interpretation, and writing of the manuscript; N.A-.P.: data interpretation and writing of the manuscript; N.J.: revised the manuscript critically; R.A.: revised the manuscript critically; N.L.D.: revised the manuscript critically; J.F.T.: bioinformatic data retrieval; G.N.G.: data interpretation and writing of the manuscript; P.W.F.: conceived the study design, data interpretation, and writing of the manuscript.

## Funding

## Competing interests
The authors declare no competing interests.

## Ethics approval
This study was conducted using publicly available data and therefore did not require ethical approval.
