## [Peer Review File · Nature Communications]

Reviewers' comments:

Reviewer #1 (Remarks to the Author):

The aim of this study is to investigate the relationship between pre-diabetes and vascular complications. While much of the investigation is conventional, there are aspects that are quite unconventional. I do not have the clinical knowledge to speak definitively, but several of these aspects (particularly relating to the genetic analysis) raise concerns.

1) About the observational analysis, while the data are impressive in size and scale, it's not clear where the data come from. What were the inclusion criteria for the various studies? Are these samples from the general population? Or ascertained samples? What do the estimates represent? Were diabetics included in the analyses? How was "pre-diabetes" defined? How was it treated in the analysis? As a binary variable (yes/no)? Or as a continuous scale? Did you take into account how long people had "pre-diabetes" for? There also appear to be many studies included multiple times (eg Deedwania 2013 for HF, Angina and MI - these outcomes are not independent). And the outcome is not consistent between the studies. For me, if it is going to be included in the manuscript, more attention needs to be paid to the systematic review / meta-analysis - there is a large amount of variability between studies and estimates that could be treated with more care.

2) About the genetic analysis, it seems strange to me conceptually to think of genetic variants that influence HbA1c, but do not influence T2D. One of the accepted definitions of T2D is a HbA1c level above a given threshold. As I said, I don't have the biological understanding to appreciate why these variants would influence glycaemic control and not risk of T2D, but the fact that they do this suggests that they influence HbA1c in an unusual way - via an unusual mechanism. I hope someone with more knowledge than me can explain why these genetic variants differ from other glycaemic-related variants. My concern is that these variants do not paint a complete picture of the relationship between glycaemic traits, and indeed that the picture arising from considering these variants may be unrepresentative of the relationship.

3) About the framing of the genetic analysis - in Mendelian randomization, the unit of investigation is not the individual person, but instead the genetic variant. So by restricting the analysis to this unusual set of genetic variants, the investigators are not considering the effect of glycaemic traits in non-diabetic study participants. They are restricting the analysis to variants that influence glycaemic traits but not T2D (although see point 4 below). So I find the whole framing of the paper a bit off - it's not that the results hold for individuals who are pre-diabetic (as the analyses weren't restricted to pre-diabetes). It's that they hold for a subset of glycaemic-related mechanisms - but it's not clear to me which.

4) While the individual variants are not associated with T2D individually at $p < 0.05$ - there is a potential for a false negative findings here. It would be interesting to combine the genetic variants into a gene score. I'd be surprised if the score does not correlate with T2D risk. This is important, as it means that the results could still be influenced by diabetes (not pre-diabetes).

5) Additionally, the authors talk about T2D and pre-diabetes as if these are two phenomena that can be separated. Whereas in truth, the line between the two is fuzzy.

Overall, the manuscript has the sense of being written by someone who does not understand the applied context of the analyses they are performing - the numbers are (mostly) fine, but the interpretation of what the analyses are estimating and why these analyses were chosen is somewhat lacking.

I would advise the authors to think carefully about what question they want to address (what is the causal question of interest?), potentially in the form of a hypothetical intervention or even a trial. And then think how they can most closely approximate that trial. At the moment, it seems that they are just analysing whatever data is in front of them rather than thinking critically about the analysis question.

Minor points:

6) Abstract: Would like to see estimates (plus expression of uncertainty). This is important for judging null results.

7) Would like to see a bit more detail on the MR analyses - for example a scatter plot of the genetic associations with risk factor and outcome.

Reviewer #2 (Remarks to the Author):

The aim was to evaluate the association between prediabetes with coronary artery disease (CAD), stroke and chronic kidney disease using Mendelian Randomisation (MR). The MR indicated that in European populations, fasting blood glucose values in the prediabetic range were only associated with CAD.

This is an important study. However, there are a number of critical comments.

Major points:

1. There is no universal accepted definition of prediabetes (e.g. WHO vs ADA). This was reflected in the meta-analysis of observational studies (Figures 1, 4,5). However, in the MR analysis, results for 2h glucose (OGTT) was not included. IGT has been previously related to cardiovascular risk. The lack of data on 2h glucose is a major drawback, which needs to be discussed.

2.The “absence of statistical evidence is evidence of absence”-fallacy: The association between prediabetes and outcomes that are not statistically significant is assumed to be zero.

In the MR analyses, increased HbA1c was associated with CAD after excluding SNPs associated with fasting glucose or after excluding erythrocytic SNPs (OR=1.2 and OR=1.21, respectively), although not statistically significant. This information should be given in the abstract and summary (discussion).

3.The statistical methods for the MR analysis should be explained in more detail. In particular, “directional horizontal pleiotrophy” and the Egger intercept needs to be explained.

Minor points:

1.Title: “Precision prevention” is no topic of the project and could be deleted from the title.

2.Abstract: There are a number of errors and faults.

The IDF estimate (352 million) is for IGT (not total prediabetes). Well-designed observational studies yield similar results than RCTs! Abbreviations (CAD, CKD) need to be introduced first.

The conclusions (...to begin prevention...) is not justified by the data. Use another conclusion.

3.Introduction (line 30): impaired HbA1c is not used by the ADA.

4.Results: A general problem with the data analysis is that up to one-third of people with prediabetes will develop type 2 diabetes during follow-up. This is important because in the Da Qing study the risk of mortality (mostly CAD) increased largely only in those people with IGT who developed type 2 diabetes during follow-up (Gong et al. Diabetes Care 2016).

5.Discussion: The discussion should be more focused on the study results. As an example, the second paragraph on page 11 should be completely deleted (not related to the study).

6.Methods: The authors only used PubMed, which is a drawback. Studies listed in Embase only are missing. Although the authors used the somewhat outdated NOS for bias assessment. It is unclear how this quality assessment was included in the analyses.

7.MR analyses were restricted to participants of European descent. How was this restriction carried out in multi-ethnic studies? This is an important information which needs to be included in various parts of the manuscript (starting with the title!).

Reviewer #3 (Remarks to the Author):

This question is of interest because of the debate about the independent role of higher glucose levels in the non-diabetic range and their role in disease.

Main concern: how well have the authors separate “prediabetes” from diabetes? there is too little information on the SNPs used in the MR analyses. Whilst the authors state that they used 44 independent variants associated with FG at 5×10^{-8} but $p < 0.05$ with T2D in the recent GWAS of these traits, they provide no details of how these SNPs associate with type 2 diabetes (details of FG and outcome data are in ST5). To really show that they have a genetic instrument for prediabetes that is separate from T2D, they should show the individual SNP associations with T2D and perform a two sample MR of prediabetes to T2D to show that it is flat. I would also want to see that it is flat with other CVD risk factors such as triglycerides and BMI and other measures of adiposity that influence heart disease. My suspicion is that some variants could primarily affect adiposity and then lipids and FG, meaning they are poor instruments for FG. They should also confirm the association with FG by giving by showing the FG MR with FG as a positive control – from ST5 it looks like almost half the SNPs used have a negative effect on FG and half a positive although this may be due to some confusing “effect allele” nomenclature in ST5. Whilst there are a handful of variants known to have stronger effects on FG than T2D (relative to other T2D and FG variants) these are few in number I believe. These variants are also better described as variants that alter FG levels across the full non-diabetic range.

Minor points to consider:

1. The results come before the methods. It would therefore help the reader to know a little more about the genetic variants selected to attempt to separate pre from full diabetes in the results. Perhaps this is a factor in the heterogeneity found in the estimates from the observational studies ?
2. Survival bias. Older people with prediabetes may be enriched for independent factors that protect from heart disease. This can be checked relatively easily with meta-regression or stratification by age of individuals or average of cohorts. Are the effects similar in younger compared to older groups ?

Tim Frayling

Reviewer #4 (Remarks to the Author):

The major implication of this finding is that the prevention of diabetes-related CAD should start well before diabetes is manifest.

The conclusion is probably incorrect or at least unsupported by the study. "Specifically, therapeutic interventions should focus on lowering blood glucose within the normal range for those with prediabetes and preventing those with normal glucose levels from progressing to prediabetes." It is unlikely that the only metabolic derangement of pre-diabetes is elevated glucose, so the 13% incremental risk may be from the metabolic syndrome association of abnormal blood glucose (elevations in inflammation, lipid and blood pressure abnormalities) that all may be subclinical. The new cholesterol guidelines (both ESC in 2019 and AHA/ACC in 2018) call out metabolic syndrome as a risk marker, acknowledging that the treatment of this condition is not lowering blood glucose but being more aggressive with lipid lowering and other preventive measures. Several trials of diabetes lowering medications in pre-diabetes did not show benefit (ACT NOW, DREAM) except for lifestyle changes with diabetes prevention program. The guidelines and other studies may warrant mention in discussion.

REBUTTAL: NCOMMS-19-28408

Reviewer #1 (Remarks to the Author):

The aim of this study is to investigate the relationship between pre-diabetes and vascular complications. While much of the investigation is conventional, there are aspects that are quite unconventional. I do not have the clinical knowledge to speak definitively, but several of these aspects (particularly relating to the genetic analysis) raise concerns.

Review comment 1.1: About the observational analysis, while the data are impressive in size and scale, it's not clear where the data come from. What were the inclusion criteria for the various studies? Are these samples from the general population? Or ascertained samples? What do the estimates represent? Were diabetics included in the analyses? How was "pre-diabetes" defined? How was it treated in the analysis? As a binary variable (yes/no)? Or as a continuous scale? Did you take into account how long people had "pre-diabetes" for? There also appear to be many studies included multiple times (eg Deedwania 2013 for HF, Angina and MI - these outcomes are not independent). And the outcome is not consistent between the studies. For me, if it is going to be included in the manuscript, more attention needs to be paid to the systematic review / meta-analysis - there is a large amount of variability between studies and estimates that could be treated with more care.

Response 1.1: We provided a detailed overview of how the studies were identified in the supplementary materials. Specifically:

"... it's not clear where the data come from. What were the inclusion criteria for the various studies?"

The inclusion criteria are described in the section: "Observational data meta-analysis" (main text) and "S1 Supplementary table 1", where we included cohorts with IFG and IGT defined either by ADA and/or WHO criteria as exposures at baseline and outcomes at follow-up.

"...Are these samples from the general population? Or ascertained samples?"

As described in section: "Observational data meta-analysis", the data are derived from epidemiological studies including clinical and general population samples. Further details are also included in Supplementary Table 2.

"...What do the estimates represent?"

We have now added the units for the estimates to the relevant figures.

"...Were diabetics included in the analyses?"

It is likely that people who developed diabetes between baseline and the occurrence of a cardiovascular event are included in these cohorts. This is one of the main weaknesses of published observational analyses of prediabetes and cardiovascular disease, and is thus one of the motivations for the current causal inference analyses.

“...How was "pre-diabetes" defined? How was it treated in the analysis? As a binary variable (yes/no)? Or as a continuous scale?”

Prediabetes is a binary classification according to the most widely accepted standards of care. Thus, prediabetes was defined by the authors of the original publications using either WHO or ADA criteria. This is described in the Methods section titled “Observational data meta-analysis”.

Did you take into account how long people had "pre-diabetes" for?

Indeed, the different durations of follow-up were considered in the analyses as a source of heterogeneity by performing subgroup analyses where studies were categorized into those with ‘short’ and ‘long’ follow-up durations by stratifying below or above the mean follow-up duration (9.6 yrs). Nevertheless, there was no statistically significant evidence of heterogeneity and the results were essentially the same within the two categories (see “Subgroup analyses”).

There also appear to be many studies included multiple times (eg Deedwania 2013 for HF, Angina and MI - these outcomes are not independent).

It is the case that in some instances the same study is used in different analyses. This includes where association statistics were retrieved by strata of sex or ethnicity. For the example given by the Reviewer, we removed the related outcomes and retained myocardial infarction. Meta-analyses for CHD and stroke and their respective subgroup analyses were obtained after removing potential overlapping cases.

And the outcome is not consistent between the studies. For me, if it is going to be included in the manuscript, more attention needs to be paid to the systematic review / meta-analysis - there is a large amount of variability between studies and estimates that could be treated with more care.

We appreciate the comment. To clarify, our outcome ‘coronary heart disease’ encompasses several conditions, i.e. myocardial infarction, congestive heart failure, angina pectoris. Thus, all studies reporting any of these outcomes (see Supplementary table 2) were considered suitable for the synthesis. Thereafter, any variability was assessed by common measures of heterogeneity (Cochran’s Q and Higgins’ I squared).

Review comment 1.2: About the genetic analysis, it seems strange to me conceptually to think of genetic variants that influence HbA1c, but do not influence T2D. One of the accepted definitions of T2D is a HbA1c level above a given threshold. As I said, I don't have the biological understanding to appreciate why these variants would influence glycaemic control and not risk of T2D, but the fact that they do this suggests that they influence HbA1c in an unusual way - via an unusual mechanism. I hope someone with more knowledge than me can explain why these genetic variants differ from other glycaemic-related variants. My concern is that these variants do not paint a complete picture of the relationship between glycaemic traits, and indeed that the picture arising from considering these variants may be unrepresentative of the relationship.

Response 1.2: It may help the Reviewer to consider the well-established examples of gene variants that affect the set-point for a trait but which do not cause a progressive change in the trait. An example of this in relation to glucose and HbA1c variation are mutations in the *GCK* gene that cause a type of maturity onset diabetes of the young (MODY), where blood glucose and HbA1c remain moderately elevated across the life-course, yet generally do not increase; accordingly, GCK MODY is generally perceived as a benign condition. Similarly, the gene variants we have studied predispose to subtle variations in blood glucose below the diabetic threshold. It is likely that these variants, like those in *GCK*, just don't drive a progressive deterioration in glucose over time to the point where diabetes occurs. Nevertheless, they are robust markers of non-diabetic glucose variation, which is the exposure of interest in these analyses.

Review comment 1.3: About the framing of the genetic analysis - in Mendelian randomization, the unit of investigation is not the individual person, but instead the genetic variant. So by restricting the analysis to this unusual set of genetic variants, the investigators are not considering the effect of glycaemic traits in non-diabetic study participants. There are restricting the analysis to variants that influence glycaemic traits but not T2D (although see point 4 below). So I find the whole framing of the paper a bit off - it's not that the results hold for individuals who are pre-diabetics (as the analyses weren't restricted to pre-diabetes). It's that they hold for a subset of glycaemic-related mechanisms - but it's not clear to me which.

Response 1.3: In MR analyses, the genetic instrument is considered a proxy for an exposure, which is, in this case, non-diabetic variations in blood glucose. To minimize risk that the genetic instrument is characterizing a specific molecular process, multiple variants that act across multiple pathways are used. Whether the glucose is synthesized by the liver or kidneys or any other organ, this should not affect its effect on the vasculature. With respect to the framing of the paper, the Reviewer is correct that the results do not directly pertain to one individual; this is also true of any other epidemiological analyses, where the effect estimates represented are derived as averages for the entire population with varying degrees of relevance for any individual participant.

Review comment 1.4: While the individual variants are not associated with T2D individually at $p < 0.05$ - there is a potential for a false negative findings here. It would be interesting to combine the genetic variants into a gene score. I'd be surprised if the score does not correlate with T2D risk. This is important, as it means that the results could still be influenced by diabetes (not pre-diabetes).

Response 1.4:

We performed two confirmatory analyses in this regard. First, we did a Mendelian randomization analysis on the association between the selected prediabetes SNPs (that were not associated with T2D at nominal significance) and the risk of T2D. There was no significant relationship though there was modest evidence of horizontal pleiotropy. This has been detailed in **Response 3.1**.

Secondly, as requested by the reviewer, we combined the 37 SNPs that passed quality control (QC) in the above analysis into a weighted genetic risk score (GRS) and tested the relationship between the GRS and risk of T2D in individual-level data from a large cohort, the UK Biobank (N = 398,810). From these analyses, the prediabetes GRS was not significantly associated with the risk of T2D, OR = 0.91, 95% CI: 0.79, 1.04, $p = 0.15$; for a logistic regression model adjusted for BMI only and OR = 0.90, 95% CI: 0.79, 1.02, $p = 0.10$; for a multivariable logistic regression model adjusted additionally for age, sex, batch effect and the

first ten genetic principal components. Additional adjustment for behavioural factors did not change the results.

Review comment 1.5: Additionally, the authors talk about T2D and pre-diabetes as if these are two phenomena that can be separated. Whereas in truth, the line between the two is fuzzy.

Response 1.5: The point raised by the Reviewer is at the heart of many debates about “prediabetes” and underlies some of the motivation for this paper. Currently, it is not established if prediabetes should be considered part of a continuum of risk or a separate category. Nevertheless, the major standards of care imply that it is indeed a separate category and it is for this reason that we refer to prediabetes in this way in our paper.

Review comment 1.6: Overall, the manuscript has the sense of being written by someone who does not understand the applied context of the analyses they are performing - the numbers are (mostly) fine, but the interpretation of what the analyses are estimating and why these analyses were chosen is somewhat lacking.

Response 1.6: The Reviewer’s opinion is noted

Review comment 1.7: I would advise the authors to think carefully about what question they want to address (what is the causal question of interest?), potentially in the form of a hypothetical intervention or even a trial. And then think how they can most closely approximate that trial. At the moment, it seems that they are just analysing whatever data is in front of them rather than thinking critically about the analysis question.

Response 1.7: The question is simply whether variations in non-diabetic glucose are causally related to cardiovascular complications. This is stated in the last sentence of the introduction.

Minor points:

Review comment 1.8: Abstract: Would like to see estimates (plus expression of uncertainty). This is important for judging null results.

Response 1.8: Now added to the Abstract

Review comment 1.9: Would like to see a bit more detail on the MR analyses - for example a scatter plot of the genetic associations with risk factor and outcome.

Response 1.9: We have included a scatter plot of the association between fasting glucose (representing prediabetes) and CAD. This has also been included in the main text.

Figure 1. Scatter plot of the genetic associations between FG and CAD. The effect sizes have been orientated to take a positive sign in relation to the exposure. The blue line represents the IVW regression for Mendelian randomization causal relationship between the exposure and outcome. In this case, there is a positive causal relationship.

Reviewer #2 (Remarks to the Author):

The aim was to evaluate the association between prediabetes with coronary artery disease (CAD), stroke and chronic kidney disease using Mendelian Randomisation (MR). The MR indicated that in European populations, fasting blood glucose values in the prediabetic range were only associated with CAD.

This is an important study. However, there are a number of critical comments.

Major points:

Review comment 2.1: There is no universal accepted definition of prediabetes (e.g. WHO vs ADA). This was reflected in the meta-analysis of observational studies (Figures 1, 4,5). However, in the MR analysis, results for 2h glucose (OGTT) was not included. IGT has been previously related to cardiovascular risk. The lack of data on 2h glucose is a major drawback, which needs to be discussed.

Response 2.1: We did not include 2hr glucose in the MR analysis, as the number of SNPs available for this analysis is few (after quality control, n=2). However, to address the Reviewer’s point to the best extent possible, we have undertaken an MR analysis using these two 2hr glucose SNPs. This analysis showed the following:

2.1a: Using SNPs that are genome-wide significantly associated with 2hr glucose and not associated with T2D at nominal significance (n = 2), the causal association between 2 hr glucose and risk of cardiovascular disease was not statistically significant (OR = 1.10, 95%

CI: 0.95, 1.27, $p = 0.22$ using IVW method). There was no evidence of heterogeneity, $Q = 34\%$, $p = 0.56$, although this was expected due to the very low number of variants. It was not possible to run MR-egger analyses or weighted median MR because these methods require more than two genetic variants for analysis.

2.1b: We also assessed the causal association between 2hr glucose and risk of CAD using 2hr glucose SNPs that were genome-wide significant, irrespective of whether they were or were not associated with T2D. After QC, the resultant number was seven uncorrelated SNPs. There was no significant association detected; however, there was evidence of horizontal pleiotropy with robust and penalized MR-Egger regression methods (See table below).

Table 1. Causal association between 2-hr glucose and the risk of CAD, $n = 7$ SNPs

Method	OR	Lower 95% CI	Upper 95% CI	P-value
Simple median	0.976	0.844	1.130	0.74
Weighted median	0.985	0.862	1.130	0.83
Penalized weighted median	0.976	0.845	1.130	0.74
IVW	1.000	0.865	1.170	0.96
Penalized IVW	0.968	0.846	1.110	0.64
Robust IVW	0.996	0.852	1.160	0.96
Penalized robust IVW	0.967	0.864	1.080	0.56
MR-Egger	0.604	0.280	1.300	0.20
(intercept)	1.050	0.977	1.120	0.19
Penalized MR-Egger	0.603	0.281	1.290	0.20
(intercept)	1.050	0.978	1.120	0.18
Robust MR-Egger	0.595	0.404	0.877	0.01
(intercept)	1.050	1.010	1.080	0.00
Penalized robust MR-Egger	0.595	0.405	0.874	0.01
(intercept)	1.050	1.010	1.080	0.00

Review comment 2.2: The “absence of statistical evidence is evidence of absence”-fallacy: The association between prediabetes and outcomes that are not statistically significant is assumed to be zero.

Response 2.2: We agree with the Reviewer, which is why we present 95% CIs for the effect estimates, as this allows the reader to judge for themselves whether the absence of a statistical association is meaningful.

Review comment 2.3: In the MR analyses, increased HbA1c was associated with CAD after excluding SNPs associated with fasting glucose or after excluding erythrocytic SNPs (OR=1.2 and OR=1.21, respectively), although not statistically significant. This information should be given in the abstract and summary (discussion).

Response 2.3: We have added the following information in the abstract as requested (page 1, lines 16 – 18 in the manuscript):

“HbA1c was associated with risk of CAD (OR=1.25 (95% CI: 1.01, 1.57)) with no evidence of directional horizontal pleiotropy ($P_{\text{Egger Intercept}} = 0.25$). This association was not significant

after excluding SNPs that were also associated with FG (OR=1.20 (95% CI: 0.91, 1.59)); after excluding only erythrocytic SNPs (OR=1.21 (95% CI: 0.96,1.52)); or excluding both, (OR=1.11 (95% CI: 0.83, 1.48)).”

Review comment 2.4: The statistical methods for the MR analysis should be explained in more detail. In particular, “directional horizontal pleiotropy” and the Egger intercept needs to be explained.

Response 2.4: We have provided further explanation as follows (page 21, lines 416 - 431 in the manuscript):

“MR-Egger is used to test for directional horizontal pleiotropy, a violation of the instrumental variable assumption where the effect of the instrumental variable on the outcome is via another pathway other than presumed. MR-Egger tests for violation of IV assumptions and bias in the inverse variance-weighted (IVW) methods and includes the intercept as part of the regression (unlike IVW, where the intercept is forced to zero)²⁹. The resulting coefficient, therefore, provides an asymptotically consistent estimate of the causal effect, even if all variants are pleiotropic with the outcome³⁰. This holds when the InSIDE (Instrument Strength Independent of Direct Effect) assumption is true, i.e. the instrument strength is independent of its pleiotropic effect. When this criterion is met, MR-Egger provides an unbiased assessment of the association between the exposure and outcome, providing the intercept, which provides the average pleiotropic effect, does not significantly differ from the null. When the intercept is significantly different from the null, it represents an estimate of the directional horizontal pleiotropic effect of the genetic variants³⁰. The median-weighted method provides a reliable estimate of the causal association between exposure and outcome when at least half of the instrumental variables are valid³¹.”

Minor points:

Review comment 2.5: Title: “Precision prevention” is no topic of the project and could be deleted from the title.

Response 2.5: We have removed this from the title

Review comment 2.6: Abstract: There are a number of errors and faults. The IDF estimate (352 million) is for IGT (not total prediabetes). Well-designed observational studies yield similar results than RCTs! Abbreviations (CAD, CKD) need to be introduced first.

The conclusions (...to begin prevention...) is not justified by the data. Use another conclusion.

Response 2.6: We have modified the paper according to these suggestions

Review comment 2.7: Introduction (line 30): impaired HbA1c is not used by the ADA.

Response 2.7: We have changed this accordingly

Review comment 2.8: Results: A general problem with the data analysis is that up to one-third of people with prediabetes will develop type 2 diabetes during follow-up. This is

important because in the Da Qing study the risk of mortality (mostly CAD) increased largely only in those people with IGT who developed type 2 diabetes during follow-up (Gong et al. Diabetes Care 2016).

Response 2.8: It is unclear whether the Reviewer is requesting any modifications to the paper in response to this point. Please clarify.

Review comment 2.9: Discussion: The discussion should be more focused on the study results. As an example, the second paragraph on page 11 should be completely deleted (not related to the study).

Response 2.9: We embarked on this analysis following a biomarker approval guidance meeting with the European Medicines Agency (EMA). A point that the EMA raised is that it is currently challenging to consider therapeutics for approval that are intended for use in prediabetes, because there is an absence of evidence that intervening in prediabetes directly reduces cardiovascular risk. This view is also shared by many in the major diabetes drug companies. Thus, the paragraph the Reviewer asks us to remove speaks to the underlying motivation to conduct this study. Accordingly, we feel it's appropriate to leave it in.

Review comment 2.10: Methods: The authors only used PubMed, which is a drawback. Studies listed in Embase only are missing. Although the authors used the somewhat outdated NOS for bias assessment. It is unclear how this quality assessment was included in the analyses.

Response 2.10:

Although PubMed is generally considered a comprehensive catalogue of published biomedical science literature, it is likely that studies exist that are not included in PubMed. If the Reviewer agrees that PubMed is a comprehensive, albeit not complete, catalogue, then a key question is whether by using PubMed we are able to obtain an unbiased set of results. To check this, we tested for publication bias using funnel plots and the Egger's test, which suggested that the results are not prone to publication bias (coefficient: 0.17 and $p=0.53$).

We then used the Newcastle-Ottawa tool for quality assessment. We tested heterogeneity in two subgroups to compare (NOS average ≤ 7 and >7 score). The subgroup test suggested a statistically significant subgroup effect ($p < 0.05$), meaning that the quality of a study may bias the estimated risk of CAD, such that CAD risk was inflated in studies with a low quality score (see Table below).

Newcastle-Ottawas scale	# studies	CAD (RR; 95% CI)			P = 0.061	# studies	CKD (RR; 95% CI)			P = 0.550	# studies	Stroke (RR; 95% CI)			P = 0.567
≤ 7	30	1.197	1.127	1.270		5	1.004	0.983	1.026		11	1.068	0.934	1.221	
> 7	9	1.103	1.039	1.172		6	1.032	0.945	1.128		12	1.116	1.041	1.196	

Review comment 2.11: MR analyses were restricted to participants of European descent. How was this restriction carried out in multi-ethnic studies? This is an important information which needs to be included in various parts of the manuscript (starting with the title!).

Response 2.11: We have clarified that the MR analysis was conducted in people of predominantly European ancestry and that it is not known if these results transfer to other ethnic groups.

Reviewer #3 (Remarks to the Author):

This question is of interest because of the debate about the independent role of higher glucose levels in the non-diabetic range and their role in disease.

Review comment 3.1: Main concern: how well have the authors separate “prediabetes” from diabetes? there is too little information on the SNPs used in the MR analyses. Whilst the authors state that they used 44 independent variants associated with FG at 5×10^{-8} but $p < 0.05$ with T2D in the recent GWAS of these traits, they provide no details of how these SNPs associate with type 2 diabetes (details of FG and outcome data are in ST5). To really show that they have a genetic instrument for prediabetes that is separate from T2D, they should show the individual SNP associations with T2D and perform a two sample MR of prediabetes to T2D to show that it is flat. I would also want to see that it is flat with other CVD risk factors such as triglycerides and BMI and other measures of adiposity that influence heart disease. My suspicion is that some variants could primarily affect adiposity and then lipids and FG, meaning they are poor instruments for FG. They should also confirm the association with FG by giving by showing the FG MR with FG as a positive control – from ST5 it looks like almost half the SNPs used have a negative effect on FG and half a positive although this may be due to some confusing “effect allele” nomenclature in ST5. Whilst there are a handful of variants known to have stronger effects on FG than T2D (relative to other T2D and FG variants) these are few in number I believe. These variants are also better described as variants that alter FG levels across the full non-diabetic range.

Response 3.1: We have added data on the analyses suggested by the Reviewer as detailed below.

3.1a: Association between the 44 FG SNPs and T2D to demonstrate a “flat” relationship with diabetes.

In these analyses, the association between prediabetes-only SNPs and risk of T2D was not significant, in the words of Reviewer, it was flat. There is however modest evidence of horizontal pleiotropy with penalized MR-egger regression methods. The resultant number of SNPs was 37, as seven were excluded for being palindromic with intermediate minor allele frequencies during harmonization of the exposure and outcome datasets.

Table 2. Causal association between prediabetes only and risk of T2D. n = 37 SNPs

Method	OR	Lower 95% CI	Upper 95% CI	P-value
Simple median	1.430	1.000	2.03	0.05
Weighted median	1.040	0.766	1.40	0.81
Penalized weighted median	1.040	0.766	1.40	0.81
IVW	1.110	0.898	1.37	0.34
Penalized IVW	1.120	0.904	1.38	0.30

Table 2. Causal association between prediabetes only and risk of T2D. n = 37 SNPs

Method	OR	Lower 95% CI	Upper 95% CI	P-value
Robust IVW	1.130	0.842	1.50	0.42
Penalized robust IVW	1.140	0.854	1.51	0.38
MR-Egger	0.767	0.515	1.14	0.19
(intercept)	1.010	1.000	1.02	0.03
Penalized MR-Egger	0.767	0.515	1.14	0.19
(intercept)	1.010	1.000	1.02	0.03
Robust MR-Egger	0.774	0.569	1.05	0.10
(intercept)	1.010	1.000	1.02	0.00
Penalized robust MR-Egger	0.774	0.569	1.05	0.10
(intercept)	1.010	1.000	1.02	0.00

We additionally assessed the association between all genome-wide significant FG SNPs and the risk of T2D. After QC there was a total of 67 eligible SNPs. As expected, FG was strongly causally associated with T2D using all methods. However, there was also a high degree of horizontal pleiotropy, $P_{\text{Egger Intercept}} = 4.82e-03$. Table 3 below shows the results.

Table 3. Causal association between fasting glucose (all GWA significant) and risk of T2D. n = 67

Method	OR	Lower 95% CI	Upper 95% CI	P-value
Simple median	3.290	2.420	4.47	3.26e-14
Weighted median	3.190	2.350	4.32	7.42e-14
Penalized weighted median	3.220	2.380	4.36	4.11e-14
IVW	4.310	2.380	7.77	1.28e-06
Penalized IVW	3.460	2.800	4.29	5.73e-30
Robust IVW	2.990	1.810	4.96	2.05e-05
Penalized robust IVW	3.370	2.650	4.27	1.40e-23
MR-Egger	0.887	0.258	3.05	8.49e-01
(intercept)	1.050	1.010	1.08	4.82e-03
Penalized MR-Egger	1.210	0.740	1.97	4.50e-01
(intercept)	1.030	1.020	1.04	1.60e-06
Robust MR-Egger	1.080	0.420	2.79	8.70e-01
(intercept)	1.030	1.000	1.05	2.79e-02
Penalized robust MR-Egger	1.530	0.472	4.98	4.77e-01
(intercept)	1.020	0.996	1.05	9.22e-02

3.1b: Association with other measures associated with risk of CVD.

We tested the causal association with several additional risk factors for CAD (LDL cholesterol, HDL cholesterol, Total Cholesterol, Triglycerides and BMI). Only triglyceride levels had a weak causal relationship with genetically determined fasting glucose and there was modest evidence of directional horizontal pleiotropy for triglycerides.

“They should also confirm the association with FG by giving by showing the FG MR with FG as a positive control – from ST5 it looks like almost half the SNPs used have a negative effect

on FG and half a positive although this may be due to some confusing “effect allele” nomenclature in ST5”

3.1c: Yes, it is indeed true that we have SNPs that are both directly and inversely related to genetically determined fasting glucose levels in the prediabetes instruments. This is expected. It however does not necessarily translate into the same proportion of positive and negative SNPs in the outcome SNPs which are integral in the two sample MR analysis. As requested by reviewer, we checked the association using fasting glucose SNPs with positive effects only and with negative effects only. In both cases, the association between prediabetes and CAD was statistically significant using robust IVW (OR = 1.24 , 95% CI:1.090, 1.42, p = 0.001, $P_{EggerIntercept} = 0.95$ for positive only SNPs and OR = 1.23, 95% CI: 1.04, 1.46; p = 0.01; $P_{EggerIntercept} = 0.40$, for negative only SNPs) with no evidence of horizontal pleiotropy. The effect size did not differ substantially. However, this separation of effect sizes gives a biased instrument that would fail to capture all related SNPs in the outcome trait.

Minor points to consider:

Review comment 3.2: The results come before the methods. It would therefore help the reader to know a little more about the genetic variants selected to attempt to separate pre from full diabetes in the results. Perhaps this is a factor in the heterogeneity found in the estimates from the observational studies ?

Response 3.2: We have added more information about the selected SNPs in the Results section (page 5). On the results coming before the methods, we have followed the journal’s guidelines on manuscript formatting.

Review comment 3.3: Survival bias. Older people with prediabetes may be enriched for independent factors that protect from heart disease. This can be checked relatively easily with meta-regression or stratification by age of individuals or average of cohorts. Are the effects similar in younger compared to older groups ?

Response 3.3:

Following the Reviewer’s recommendation, we stratified the data above and below the average age of the collection of cohorts (57 y); there was no evidence for quantitative or qualitative heterogeneity between the two subgroups for the outcome of CAD (see Subgroup analysis in supplementary material):

Means age	# studies	CAD			P=0.396	# studies	CKD			P= 0.244	# studies	Stroke			P= 0.203
<57	22	1.164	1.105	1.225		8	1.013	0.989	1.037		13	1.126	1.052	1.205	
≥ 57	17	1.118	1.037	1.207		3	1.006	0.985	1.026		10	1.013	0.874	1.174	

Reviewer #4 (Remarks to the Author):

The major implication of this finding is that the prevention of diabetes-related CAD should start well before diabetes is manifest.

Review comment 4.1: The conclusion is probably incorrect or at least unsupported by the study. “Specifically, therapeutic interventions should focus on lowering blood glucose within the normal range for those with prediabetes and preventing those with normal glucose levels from progressing to prediabetes.” It is unlikely that the only metabolic derangement of pre-diabetes is elevated glucose, so the 13% incremental risk may be from the metabolic syndrome association of abnormal blood glucose (elevations in inflammation, lipid and blood pressure abnormalities) that all may be subclinical. The new cholesterol guidelines (both ESC in 2019 and AHA/ACC in 2018) call out metabolic syndrome as a risk marker, acknowledging that the treatment of this condition is not lowering blood glucose but being more aggressive with lipid lowering and other preventive measures. Several trials of diabetes lowering medications in pre-diabetes did not show benefit (ACT NOW, DREAM) except for lifestyle changes with diabetes prevention program.

Response 4.1: We agree that there are many other factors that co-segregate with blood glucose that affect cardiovascular risk. Indeed, this is why we elected to use MR for the current analyses, as this is relatively robust to the types of confounding highlighted by the Reviewer. However, there are certain types of confounding that MR is prone to, one of which is termed “horizontal pleiotropy”. Reviewer 3 (see comment 3.1) raises this possibility and asked us to undertake tests to exclude it. We have done so and report the findings of these analyses above.

Review comment 4.2: The guidelines and other studies may warrant mention in discussion.

Response 4.2: We have briefly mentioned these guidelines in the Discussion.

Tables of associations between prediabetes only and other factors associated with CAD

Table 4. Causal association between prediabetes only and LDL cholesterol levels

Method	Estimate	Lower_CI	Upper_CI	P.value
Simple median	1.000	0.904	1.11	0.96
Weighted median	1.070	0.979	1.17	0.13
Penalized weighted median	1.070	0.982	1.17	0.12
IVW	1.070	0.909	1.27	0.40
Penalized IVW	0.991	0.928	1.06	0.78
Robust IVW	0.993	0.910	1.08	0.88
Penalized robust IVW	1.010	0.922	1.11	0.80
MR-Egger	1.010	0.732	1.40	0.94
(intercept)	1.000	0.993	1.01	0.69
Penalized MR-Egger	1.060	0.942	1.20	0.32
(intercept)	0.998	0.995	1.00	0.26
Robust MR-Egger	1.070	0.970	1.19	0.17
(intercept)	0.997	0.994	1.00	0.13
Penalized robust MR-Egger	1.090	0.988	1.20	0.09
(intercept)	0.998	0.994	1.00	0.22

Table 5. Causal association between prediabetes only and HDL levels

Method	Estimate	Lower_CI	Upper_CI	P.value
Simple median	1.030	0.935	1.13	0.58
Weighted median	1.020	0.935	1.12	0.60
Penalized weighted median	1.020	0.935	1.12	0.62
IVW	1.140	0.968	1.35	0.12
Penalized IVW	1.010	0.949	1.06	0.86
Robust IVW	1.010	0.944	1.08	0.82
Penalized robust IVW	1.000	0.935	1.08	0.93
MR-Egger	0.883	0.645	1.21	0.44
(intercept)	1.010	1.000	1.02	0.06
Penalized MR-Egger	0.981	0.881	1.09	0.73
(intercept)	1.000	0.998	1.00	0.49
Robust MR-Egger	0.990	0.879	1.12	0.87
(intercept)	1.000	0.997	1.00	0.74
Penalized robust MR-Egger	0.991	0.874	1.12	0.88
(intercept)	1.000	0.998	1.00	0.68

Table 6. Causal association between prediabetes only and triglyceride levels

Method	Estimate	Lower_CI	Upper_CI	P.value
Simple median	0.897	0.809	0.996	0.04
Weighted median	0.982	0.906	1.070	0.67
Penalized weighted median	0.990	0.911	1.080	0.82
IVW	0.801	0.647	0.990	0.04
Penalized IVW	0.971	0.916	1.030	0.33
Robust IVW	0.952	0.874	1.040	0.26
Penalized robust IVW	0.979	0.933	1.030	0.37
MR-Egger	1.130	0.755	1.680	0.56
(intercept)	0.990	0.979	1.000	0.05
Penalized MR-Egger	1.100	0.976	1.250	0.11
(intercept)	0.993	0.990	0.997	0.00
Robust MR-Egger	0.993	0.878	1.120	0.90
(intercept)	0.998	0.993	1.000	0.60
Penalized robust MR-Egger	1.050	0.953	1.160	0.31
(intercept)	0.996	0.992	1.000	0.13

Table 7. Causal association between prediabetes only and total cholesterol

Method	Estimate	Lower_CI	Upper_CI	P.value
Simple median	1.00	0.907	1.10	1.00
Weighted median	1.04	0.955	1.14	0.35
Penalized weighted median	1.04	0.957	1.14	0.34
IVW	1.05	0.885	1.25	0.57
Penalized IVW	1.02	0.951	1.09	0.62
Robust IVW	1.01	0.939	1.09	0.76
Penalized robust IVW	1.02	0.956	1.09	0.55
MR-Egger	1.02	0.730	1.44	0.89
(intercept)	1.00	0.992	1.01	0.86
Penalized MR-Egger	1.03	0.913	1.17	0.60
(intercept)	1.00	0.996	1.00	0.85
Robust MR-Egger	1.02	0.906	1.15	0.75
(intercept)	1.00	0.996	1.00	0.89
Penalized robust MR-Egger	1.03	0.931	1.15	0.54
(intercept)	1.00	0.996	1.00	0.88

Table 8. Causal association between prediabetes only and BMI

Method	Estimate	Lower_CI	Upper_CI	P.value
Simple median	1.010	0.970	1.060	0.55
Weighted median	0.982	0.945	1.020	0.36
Penalized weighted median	0.982	0.945	1.020	0.36
IVW	1.000	0.927	1.090	0.94
Penalized IVW	0.983	0.955	1.010	0.26
Robust IVW	0.988	0.956	1.020	0.45
Penalized robust IVW	0.985	0.961	1.010	0.21
MR-Egger	0.914	0.789	1.060	0.23
(intercept)	1.000	0.999	1.010	0.15
Penalized MR-Egger	0.941	0.891	0.994	0.03
(intercept)	1.000	1.000	1.000	0.03
Robust MR-Egger	0.923	0.849	1.000	0.06
(intercept)	1.000	0.999	1.010	0.17
Penalized robust MR-Egger	0.949	0.899	1.000	0.06
(intercept)	1.000	0.999	1.000	0.24

Reviewers' comments:

Reviewer #1 (Remarks to the Author):

My concerns remain: the technical aspects of the data analysis (in terms of what the authors have done) seem to be performed correctly, but the motivation and interpretation of the results is deficient. Because of this, it is unclear whether the analyses performed are able to answer the authors' questions. The authors' responses demonstrate a lack of engagement with reasonable criticism from the reviewers.

1. Again, for me, there is insufficient care and attention paid to the systematic review. For example, did the studies include diabetic individuals in the analysis? The authors gave an answer to this question in the response, but I am still unclear of this. Were studies having participants with diabetics at baseline excluded? This certainly isn't clear in the main body of the manuscript reading linearly, it's not clear in the methods section, and it's not clear in the authors' response (which often was along the lines of - we've put the information in the paper, now you go and find it).

As another question, what do the estimates represent? The authors' reply: "We have now added the units for the estimates to the relevant figures." Take Figure 1. All I can see is "Relative Risk (RR) with 95% confidence intervals". Relative risk of what for what versus what? What is the comparison? Further than this, I'd be surprised if many of these estimates are relative risks - most of these would be prospective studies, so the estimates would be hazard ratios.

These are examples - I could find others.

A well-performed systematic review is a serious piece of work requiring care and attention. This is not that.

2. I remain unconvinced that considering genetic variants associated with HbA1c and not associated with T2D is a reasonable analysis strategy. I fear it could lead to bias.

For the analysis requested (does the pre-diabetes score correlate with T2D risk?), is there any association without adjustment for BMI? This seems a strange adjustment to make that could induce bias (it could be conditioning on a mediator or a collider).

3. I would appreciate a more reasoned answer to comment 1.7: "I would advise the authors to think carefully about what question they want to address (what is the causal question of interest?), potentially in the form of a hypothetical intervention or even a trial. And then think how they can most closely approximate that trial. At the moment, it seems that they are just analysing whatever data is in front of them rather than thinking critically about the analysis question."

The current answer is a pure circular logic. They define the causal effect of interest as the causal effect of interest. I would advise the authors to think more seriously about their causal question of interest through the target trial framework (<https://www.ncbi.nlm.nih.gov/pubmed/26994063>). Important questions are - how are they proposing to intervene on glucose levels? how would this influence diabetes? in whom are they proposing to intervene on glucose levels? for how long are they proposing to intervene on glucose levels? - answers to all of these questions (and more) would help clarify the target of investigation, and to what extent the Mendelian randomization investigation can estimate this target. Without knowing what causal effect the authors are targeting, it is not possible to evaluate to what extent their investigation achieves its aim.

Reviewer #2 (Remarks to the Author):

The authors have adequately addressed the critical comments raised during the review process.

However, they have stated that my following comment was not clear:

"A general problem with the data analysis is that up to onethird

of people with prediabetes will develop type 2 diabetes during follow-up. This is important because in the Da Qing study the risk of mortality (mostly CAD) increased largely only in those people with IGT who developed type 2 diabetes during follow-up (Gong et al. Diabetes Care 2016)."

The authors have used observational studies to assess the association between prediabetes and vascular complications.

A major problem of observational studies is that the transition from prediabetes to diabetes is not detected in many investigations. The Da Qing study showed that the risk of mortality (including cardiovascular mortality) largely increased in people who developed type 2 diabetes during the follow-up. A much lower risk was found in those who remained in the prediabetic state during the study period. In many studies the transition from prediabetes to type 2 diabetes is not investigated, although this seems to be a main predictor for an increased risk. My suggestion is that the authors add this point in the discussion as limitation.

Reviewer #3 (Remarks to the Author):

the authors have done a very thorough job at responding to the reviewers comments. The control analyses I suggested are very helpful (apologies for 3.1c, that is not quite what I meant i realise it was a pointless task, but most papers normalise the exposure allele to one direction).

I have only one suggestion to make it clearer to the non MR savvy reader. A "validation of genetic instruments" or similarly titled section at the start of the results would be helpful. Showing that the prediabetes and total FG genetics behave differently with FG (i dont think this is shown yet) and with T2D. I think this would strengthen the paper as it would show more clearly you have a reasonably specific instrument for FG that doesnt lead to T2D.

Tim Frayling

Reviewer #4 (Remarks to the Author):

the authors were responsive to the comments. I have no further issues

Reviewer #1 (Remarks to the Author):

Comment R1.1 My concerns remain: the technical aspects of the data analysis (in terms of what the authors have done) seem to be performed correctly, but the motivation and interpretation of the results is deficient. Because of this, it is unclear whether the analyses performed are able to answer the authors' questions. The authors' responses demonstrate a lack of engagement with reasonable criticism from the reviewers. Again, for me, there is insufficient care and attention paid to the systematic review. For example, did the studies include diabetic individuals in the analysis? The authors gave an answer to this question in the response, but I am still unclear of this.

Response R1.1: To clarify, we have made every effort to exclude any diabetic individuals from our analyses. We therefore only included studies whose populations had prediabetes ascertained through fasting glucose, HbA1c or 2hr oral glucose tolerance testing at baseline with the relevant cutoffs (e.g. ADA or WHO). Studies with individuals whose values were >7.0 mmol/L for fasting glucose, >11 mmol/L for 2hr glucose or HbA1c for >47 mmol/mol, or who self-reported diabetes or reported taking anti-diabetic medication were excluded from our analyses. This is now stated clearly in the text for readers in pages 14- 15:

“...according to the following criteria: prediabetes defined by impaired glucose tolerance (IGT), impaired fasting glucose (IFG) per WHO¹⁵ or ADA criteria and glycated haemoglobin (HbA1c) per ADA criterion¹⁶. Studies were included if participants were drawn from the general population, glycaemia was measured at baseline and the subsequent outcomes at follow-up were coronary artery disease (CAD), chronic kidney disease (CKD) or stroke, and were compared with normoglycaemic individuals...” .*“Studies with individuals known to be diagnosed with diabetes or with diabetic values at baseline or follow-up were excluded for the analysis”*.

Please note that during systematic review of the literature, we identified some studies that consider only one of IFG or IGT when including participants in their analyses (see Supplementary Material_2 Table tab: ‘Studies in the meta-analysis’, column: *prediabetes definition*). In these instances, it is possible that participants with diabetes (on the basis of the unobserved glycemic measure) may have been enrolled.

Measuring fasting glucose without an oral glucose tolerance test is sufficient for prediabetes identification/diagnosis as per ADA [1] but not for WHO [2] criteria. Therefore, we chose to perform a subgroup analysis, stratifying studies by the possibility of having inadvertently enrolled people with diabetes at baseline (‘Yes/No’) in fixed models. Results did not change from those presented from our full analysis – see Table 1 below.

Regarding the potential for development of diabetes (at follow-up), we identified some studies where it was hard to discern if individuals with i) diabetes, ii) receiving medication for diabetes or iii) missing glucose tolerance test at follow-up were actually excluded. Therefore, we added four columns to the Supplementary Material_2 Table 1 that describe each study’s quality assessment, as follows: i) Baseline Outcome Excluded (Yes/No); ii) Possibility of Enrolling patients with diabetes (Yes/No); iii) Possibility of developing diabetes during study (Yes/No); iv) number of individuals who developed diabetes (when reported). We performed the subgroup analysis on just those studies, stratified ‘Yes/No’ in fixed models. Results did not change from those presented for the full analysis – see Table 1 below.

Table 1. Subgroup Analyses of the Association between Prediabetes and Vascular outcomes

Subgroups	CAD				CKD				Stroke			
	No. studies	RR	(95% LCI, UCI)	p	No. studies	RR	(95% LCI, UCI)	p*	No. studies	RR	(95% LCI, UCI)	p
Possibility of enrolling diabetes				0.106				-				0.202
Yes	19	1.18	1.12, 1.24		11	1.01	1.00, 1.03		11	1.06	0.96, 1.17	
No	20	1.09	1.01, 1.18		-				14	1.01	0.90, 1.14	
Possibility of developing diabetes				0.239								0.478
Yes	36	1.17	1.11, 1.24		11	1.01	0.99, 1.03	-	21	1.10	1.02, 1.18	
No	3	1.11	1.04, 1.19		-	-	-	-	4	1.14	1.04, 1.26	

CAD: Coronary artery disease; CKD: Chronic kidney disease; RR: Relative risk; HR: Hazard ratio; Test for heterogeneity between sub-groups with fixed effect model (p <0.05).
 *There is no P value, because when we stratified by subgroup and test there is no comparative group, for a clear read the total (11) studies of CKD where only in one subgroup.

Comment R1.2 This [sic: “Did the studies include diabetic individuals in the analysis?”] certainly isn't clear in the main body of the manuscript reading linearly, it's not clear in the methods section, and it's not clear in the authors' response (which often was along the lines of - we've put the information in the paper, now you go and find it).

Response R1.2:

To clarify, yes, studies having participants with diabetes at baseline were excluded. Our aim was to ensure that no studies with participants with diabetes at baseline were included in our analyses. This is now stated clearly in the main body of text for readers on page 15:

“Studies with individuals known to be diagnosed with diabetes or with diabetic values at baseline or follow-up were excluded from the analysis.”

As discussed previously, all of the published studies considered in our analyses excluded individuals with diabetic fasting plasma glucose (n= 33), HbA1c (n= 5) or 2hr glucose (n= 8). According to our definition of ‘Possibility of enrolling patients with diabetes’, that is, fasting glucose only measured at baseline (i.e. not coupled with an IGT), we identified 23 (62.1%) studies possibly enrolling individuals with diabetes and 32 (86.4%) studies where participants may have developed diabetes during the course of the study.

See our previous response (R1.1), whereby results did not differ significantly from the full analyses.

Please also see our response to **Reviewer #2** below and the amendments to the limitation section (page 13).

Comment R1.3. As another question, what do the estimates represent? The authors' reply: "We have now added the units for the estimates to the relevant figures." Take Figure 1. All I can see is "Relative Risk (RR) with 95% confidence intervals". Relative risk of what for what versus what? What is the comparison? Further than this, I'd be surprised if many of these estimates are relative risks - most of these would be prospective studies, so the estimates would be hazard ratios.

Response R1.3: To clarify, the reference group to which all reported relative risks should be compared are those individuals with normal glucose levels. This is clearly stated in the Methods section page 14-15:

“Studies were included if participants were drawn from the general population, glycaemia was measured at baseline and the subsequent outcomes at follow-up were coronary artery disease (CAD), chronic kidney disease (CKD) or stroke, and were compared with the group of normoglycaemic participants”.

Similarly, in Supplementary Table 2 in the ‘Study characteristics’ tab, the footnote stipulates that the relative risk estimates are in comparison with normoglycemic individuals (i.e. estimates represent the likelihood of cardiovascular and kidney disease outcomes in individuals with prediabetes compared with those with normal glucose levels). (see Supplementary Table 2: Studies in the meta-analysis):

*“...e. ^a RR v normoglycemic; ** Men; * Women; (-) Not available. † South Asian; ‡ European.”*

Regarding concerns over hazard ratios (HR), 15 of the studies included in our analyses (40.5%) did not estimate hazard ratios. However, we stratified studies according to the type of estimate reported and no significant differences were observed (see Table 2 below). Moreover, we pooled HR-only studies, using the generic inverse-variance method for

analyzing time to event data [3] [4]; again, no differences were observed (See Figures, 1,2 and 3 - below). For the reader, we added the following sentence in page 4 (section 'Results'):

“In observational analyses, prediabetes conveyed an RR of 1.11(95%CI: 1.03, 1.18; $Q=28.5$, $P_{Qstat} = 0.23$; $I^2= 16\%$) for stroke (Fig. 3), these remained virtually unchanged in the subgroup analysis”

Table 2. Subgroup Analyses of the Association between Prediabetes and Vascular outcomes												
	CAD				CKD				Stroke			
Subgroups	No. studies	RR	(95% LCI, UCI)	p	No. studies	RR	(95% LCI, UCI)	p	No. studies	RR	(95% LCI, UCI)	p
Reported measure of association				0.260				0.389				0.615
HR	24	1.13	1.08, 1.19		5	0.99	0.95, 1.03		18	1.12	1.05, 1.19	
RR	15	1.2	1.10, 1.31		6	1.01	0.99, 1.04		7	1.08	0.93,1.24	
CAD: Coronary artery disease; CKD: Chronic kidney disease; RR: Relative risk; HR: Hazard ratio; Test for heterogeneity between sub-groups with fixed effect model ($p < 0.05$).												

Coronary artery disease (CAD)

Figure 1. Meta-analysis of the association between prediabetes and Coronary artery disease

Stroke

Figure 2. Meta-analysis of the association between prediabetes and stroke

CKD

Figure 3. Meta-analysis of the association between prediabetes and Chronic kidney disease

Comment R1.4. These are examples - I could find others. A well-performed systematic review is a serious piece of work requiring care and attention. This is not that.

Response R1.4. We followed a standardized and reproducible approach throughout, based upon PRISMA and Cochrane Handbook guidelines. In our view, by following these guidelines, the review we've performed is sufficient for the stated purpose.

Comment R1.5. I remain unconvinced that considering genetic variants associated with HbA1c and not associated with T2D is a reasonable analysis strategy. I fear it could lead to bias. For the analysis requested (does the pre-diabetes score correlate with T2D risk?), is there any association without adjustment for BMI? This seems a strange adjustment to make that could induce bias (it could be conditioning on a mediator or a collider).

Response R1.5.

We conducted the analysis using models adjusted for and unadjusted for BMI. In the latter, the prediabetes score was not associated with risk of T2D in models adjusted for 1) age, sex, batch number and the first ten principle components: OR (95% CI) = 0.95 (0.84, 1.1), p = 0.44; and 2) additionally adjusted for smoking and alcohol use: OR (95% CI) = 0.97 (0.85, 1.1), p = 0.65. By comparing both BMI-adjusted and unadjusted models, we confirmed that the association in the former is not driven by BMI and that this adjustment is unlikely to materially affect the results.

Comment R1.6. I would appreciate a more reasoned answer to comment 1.7: "I would advise the authors to think carefully about what question they want to address (what is the causal question of interest?), potentially in the form of a hypothetical intervention or even a trial. And then think how they can most closely approximate that trial. At the moment, it seems that they are just analysing whatever data is in front of them rather than thinking critically about the analysis question."

Response R1.6.

The primary research question concerns whether a causal relationship between the exposure (prediabetes) and the outcomes (a variety of vascular complications most often associated with diabetes) exists. We believe this is an interesting and relevant question, as there is much debate about whether intervening on prediabetes is likely to reduce cardiovascular risk. However, the purpose of this paper is not to speculate on or assess how different interventions might perform.

Mendelian randomization is a recently popularised adjunct to randomized controlled trials that makes use of epidemiological data for causal inference. For this approach to be robust, however, several assumptions need to be fulfilled. Using available data, we have taken great care to ensure that our genetic instruments comply with these assumptions. Indeed, as part of the revision process, we were able to update our instrumental variables with recently published variants, which improved the validity of our instruments.

Comment R1.7. The current answer is a pure circular logic. They define the causal effect of interest as the causal effect of interest. I would advise the authors to think more seriously about their causal question of interest through the target trial framework (<https://www.ncbi.nlm.nih.gov/pubmed/26994063>). Important questions are - how are they proposing to intervene on glucose levels? how would this influence diabetes? in whom are they proposing to intervene on glucose levels? for how long are they proposing to intervene on glucose levels? - answers to all of these questions (and more) would help clarify the target of investigation, and to what extent the Mendelian randomization investigation can estimate this target. Without knowing what causal effect the authors are targeting, it is not possible to evaluate to what extent their investigation achieves its aim.

Response R1.7.

To clarify, the causal target is elevated but not diabetic glucose levels (i.e., prediabetes). Though we appreciate the concept of the trial framework suggested above, this current body of work aims simply to test a hypothesis: do elevated, but not diabetic, levels of blood glucose causally affect vascular outcomes? Past observational studies indicate that people with prediabetes have elevated risk of certain vascular diseases compared to normoglycaemic individuals, but this observation may be confounded for reasons discussed at length in the paper. We undertook the MR work described herein in order to circumvent the key limitations of observational studies (confounding and reverse causality). Our findings strongly suggest that there is indeed a causal relationship between prediabetes and CAD.

Reviewer #2 (Remarks to the Author)

Comment R2. The authors have adequately addressed the critical comments raised during the review process. However, they have stated that my following comment was not clear: "A general problem with the data analysis is that up to one third of people with prediabetes will develop type 2 diabetes during follow-up. This is important because in the Da Qing study the risk of mortality (mostly CAD) increased largely only in those people with IGT who developed type 2 diabetes during follow-up (Gong et al. Diabetes Care 2016)." The authors have used observational studies to assess the association between prediabetes and vascular complications. A major problem of observational studies is that the transition from prediabetes to diabetes is not detected in many investigations. The Da Qing study showed that the risk of mortality (including cardiovascular mortality) largely increased in people who developed type 2 diabetes during the follow-up. A much lower risk was found in those who remained in the prediabetic state during the study period. In many studies the transition from prediabetes to type 2 diabetes is not investigated, although this seems to be a main predictor for an increased risk. My suggestion is that the authors add this point in the discussion as limitation.

Response R2.

We concur with Reviewer #2 and have added this point to the discussion as a limitation, as per their suggestion (page 13)

"A major limitation of observational studies is the potential that participants progress to diabetes. Therefore, we went to great lengths to identify and stratify only those studies that excluded individuals with diabetes in the analysis. Those which we deemed having the most likelihood of enrolling diabetics (i.e. those recruiting participants only with HbA1 or fasting glucose) were further stratified into a specific subgroup for re-analysis; results remained virtually unchanged (see Supplemental Material_2, Additional subgroup analysis). By no means do we claim that the observational evidence is definitive; on the contrary, this motivated us to contest these observational data and explore causality through the MR approach."

Reviewer #3 (Remarks to the Author):

Comment R3. the authors have done a very thorough job at responding to the reviewers comments. The control analyses I suggested are very helpful (apologies for 3.1c, that is not quite what I meant i realize it was a pointless task, but most papers normalise the exposure allele to one direction). I have only one suggestion to make it clearer to the non MR savvy reader. A "validation of genetic instruments" or similarly titled section at the start of the results would be helpful. Showing that the prediabetes and total FG genetics behave differently with FG (i dont think this is shown yet) and with T2D. I think this would strengthen the paper as it would show more clearly you have a reasonably specific instrument for FG that doesnt lead to T2D.

Tim Frayling

Response R3. Thanks for your latest comments. We have addressed your suggestions in the Supplementary Material_1 page 6 as follows:

S5. Validation of instruments.

We identified instruments that are associated with fasting glucose and not associated with T2D at nominal significance. To further demonstrate the suitability of our instruments, we conducted two validation analyses to rule out any other causal association with T2D. First, to demonstrate that our prediabetes-only instrument was not also causally related to T2D and thus ideal for the study question, we performed a two sample MR analysis of the relationship between prediabetes only, as defined with our instrumental variable, and the risk of T2D. After QC, the resultant number of SNPs available for analysis was 28. In these analyses, the association between prediabetes-only SNPs and risk of T2D was not statistically significant ($P>0.05$). There was however tentative evidence of horizontal pleiotropy, (Table S4.1).

Table S4.1. Causal association between prediabetes only and risk of T2D. n = 28 SNPs

Method	OR	Lower 95% CI	Upper 95% CI	P-value
Weighted median	0.98	0.82	1.14	0.786
IVW	1.02	0.90	1.16	0.761
Robust IVW	1.02	0.90	1.15	0.77
MR-Egger (intercept)	0.91 1	0.73 0.997	1.14 1.01	0.418 0.234
Robust MR-Egger (intercept)	0.91 1	0.77 0.998	1.07 1.01	0.252 0.151

Second, we investigated the relationship between all genome-wide significant FG SNPs and the risk of T2D. After QC there was a total of 74 eligible SNPs. As expected, FG was strongly causally associated with T2D using all methods. However, there was also a high degree of horizontal pleiotropy, which is consistent with the complex phenotype that is T2D (Table S4.2).

Table S4.2. Causal association between fasting glucose (all genome-wide significant) and risk of T2D. n = 74

Method	OR	Lower 95% CI	Upper 95% CI	P-value
Weighted median	1.55	1.23	1.94	1.67×10^{-4}
IVW	2.26	1.37	3.74	1.43×10^{-3}
Robust IVW	2.35	1.50	3.67	1.75×10^{-4}
MR-Egger (intercept)	0.46 1.05	0.19 1.03	1.12 1.08	0.09 5.05×10^{-5}
Robust MR-Egger (intercept)	0.96 1.03	0.45 1.01	2.03 1.04	0.91 5.54×10^{-3}

Reviewer #4 (Remarks to the Author):

The authors were responsive to the comments. I have no further issues

Response R4. Thank you

References:

- [1] Association AD (2019) 2. Classification and diagnosis of diabetes: standards of medical care in diabetes—2019. *Diabetes Care* 42(Supplement 1): S13-S28
- [2] World Health Organization (2006) Definition and diagnosis of diabetes mellitus and intermediate hyperglycemia: report of a WHO/IDF consultation. In: World Health Org
- [3] Higgins J, Green S (2011) *Cochrane handbook for systematic reviews of interventions*. Available from <http://handbook.cochrane.org>. Accessed 1/12 2017
- [4] Tierney JF, Stewart LA, Ghersi D, Burdett S, Sydes MR (2007) Practical methods for incorporating summary time-to-event data into meta-analysis. *Trials* 8: 16. Artn 16 10.1186/1745-6215-8-16
- [5] Gong Q, Zhang P, Wang J, et al. (2019) Morbidity and mortality after lifestyle intervention for people with impaired glucose tolerance: 30-year results of the Da Qing Diabetes Prevention Outcome Study. *The Lancet Diabetes & Endocrinology* 7(6): 452-461

REVIEWERS' COMMENTS:

Reviewer #1 (Remarks to the Author):

I still find the question that this study seeks to address opaque (the effect of increasing HbA1c without increasing the risk of T2D). I'm still concerned that this isn't a meaningful question, but I appreciate the authors willingness to engage. For me, concerns remain about the choice of genetic variants - I would think a better way to progress on this question would be to restrict the analysis of the outcome to non-diabetics rather than to try to find this odd group of variants that influence HbA1c but (seemingly) not T2D risk. But I feel that I've probed as much as is reasonable - what the authors have done is clear and open even if I do not fully agree that it is optimal.

Reviewer #3 (Remarks to the Author):

No further comments. the additional analyses, showing that the prediabetes/FG variants are not collectively associated with T2D are very useful. Thanks. Tim Frayling

REVIEWERS' COMMENTS:

Reviewer #1 (Remarks to the Author):

I still find the question that this study seeks to address opaque (the effect of increasing HbA1c without increasing the risk of T2D). I'm still concerned that this isn't a meaningful question, but I appreciate the authors willingness to engage. For me, concerns remain about the choice of genetic variants - I would think a better way to progress on this question would be to restrict the analysis of the outcome to non-diabetics rather than to try to find this odd group of variants that influence HbA1c but (seemingly) not T2D risk. But I feel that I've probed as much as is reasonable - what the authors have done is clear and open even if I do not fully agree that it is optimal.

Response: We thank the reviewer for their comments.

Reviewer #3 (Remarks to the Author):

No further comments. the additional analyses, showing that the prediabetes/FG variants are not collectively associated with T2D are very useful. Thanks. Tim Frayling

Response: We thank the reviewer for their comments.